# Evaluation of hydroxychloroquine or chloroquine for the prevention of COVID-19 (COPCOV): A double-blind, randomised, placebo-controlled trial

William H. K. Schilling[1,2]*, Mavuto Mukaka[1,2], James J. Callery[1,2], Martin J. Llewelyn[3,4], Cintia V. Cruz[1,2], Mehul Dhorda[1,2], Thatsanun Ngernseng[1], Naomi Waithira[1,2], Maneerat Ekkapongpisit[1], James A. Watson[2,5], Arjun Chandna[2,6], Erni J. Nelwan[7,8], Raph L. Hamers[2,9], Anthony Etyang[2,10], Mohammad Asim Beg[11], Samba Sow[12], William Yavo[13], Aurel Constant Allabi[14], Buddha Basnyat[2,15], Sanjib Kumar Sharma[16], Modupe Amofa-Sekyi[17], Paul Yonga[18], Amanda Adler[19], Prayoon Yuentrakul[1], Tanya Cope[1], Janjira Thaipadungpanit[1,20], Panuvit Rienpradub[1], Mallika Imwong[1,21], Mohammad Yazid Abdad[1,2], Stuart D. Blacksell[1,2], Joel Tarning[1,2], Frejus Faustin Goudjo[22], Ange D. Dossou[23], Abibatou Konaté-Touré[13], Serge-Brice Assi[24], Kra Ouffoué[25], Nasronudin Nasronudin[26,27], Brian Eka Rachman[26,27], Pradana Zaky Romadhon[26,27], Didi Darmahadi Dewanto[28], Made Oka Heryana[28], Theresia Novi[28], Ayodhia Pitaloka Pasaribu[29], Mutiara Mutiara[30], Miranda Putri Rahayu Nasution[30], Khairunnisa Khairunnisa[30], Fauzan Azima Dalimunthe[29], Eka Airlangga[31], Akmal Fahrezzy[31], Yanri Subronto[32], Nur Rahmi Ananda[33], Mutia Rahardjani[9], Atika Rimainar[9], Ruth Khadembu Lucinde[10], Molline Timbwa[10], Otieno Edwin Onyango[10], Clara Agutu[10], Samuel Akech[2,10], Mainga Hamaluba[2,10], Jairus Kipyego[18], Obadiah Ngachi[18], Fadima Cheick Haidara[12], Oumar Y. Traoré[12], François Diarra[12], Basudha Khanal[16], Piyush Dahal[16], Suchita Shrestha[15], Samita Rijal[15], Youssouf Kabore[34], Eric Adehossi[35], Ousmane Guindo[34], Farah Naz Qamar[36], Abdul Momin Kazi[36], Charles J. Woodrow[37,38], Steven Laird[39], Maina Cheeba[17], Helen Ayles[17,40], Phaik Yeong Cheah[1,2], Walter R. J. Taylor[1,2], Elizabeth M. Batty[1,2], Kesinee Chotivanich[1,20], Sasithon Pukrittayakamee[1,20], Weerapong Phumratanaprapin[20], Lorenz von Seidlein[1,2], Arjen Dondorp[1,2], Nicholas P. J. Day[1,2], Nicholas J. White[1,2], on behalf of the COPCOV Collaborative Group[¶]

1 Mahidol Oxford Tropical Medicine Research Unit, Faculty of Tropical Medicine, Mahidol University, Bangkok, Thailand, 2 Centre for Tropical Medicine and Global Health, Nuffield Department of Medicine, University of Oxford, Oxford, United Kingdom, 3 Department of Global Health and Infection, Brighton and Sussex Medical School, Brighton, United Kingdom, 4 Department of Microbiology and Infection, University Hospitals Sussex NHS Foundation Trust, Brighton, United Kingdom, 5 Oxford University Clinical Research Unit, Hospital for Tropical Diseases, Ho Chi Minh City, Vietnam, 6 Cambodia Oxford Medical Research Unit, Angkor Hospital for Children, Siem Reap, Cambodia, 7 Faculty of Medicine, Universitas Indonesia, Jakarta, Indonesia, 8 Division of Tropical Medicine and Infectious Diseases, Department of Internal Medicine, Dr. Cipto Mangukusumo Hospital, Jakarta, Indonesia, 9 Oxford University Clinical Research Unit Indonesia, Faculty of Medicine, Universitas Indonesia, Jakarta, Indonesia, 10 KEMRI-Wellcome Trust Research Programme, Kilifi, Kenya, 11 Department of Pathology and Laboratory Medicine, The Aga Khan University Hospital, Karachi, Pakistan, 12 Centre pour le Développement des Vaccins (CVD-Mali), Bamako, Mali, 13 Centre de Recherche et de Lutte contre le Paludisme, Institut National de Santé Publique, Abidjan, Côte d'Ivoire, 14 Faculty of Health Sciences, Laboratory of Pharmacology and Toxicology, University of Abomey-Calavi, Cotonou, Benin, 15 Oxford University Clinical Research Unit Nepal, Lalitpur, Nepal, 16 B.P. Koirala Institute of Health Sciences (BPKIHS), Dharan, Nepal, 17 Zambart, University of Zambia School of Public Health, Lusaka, Zambia, 18 Fountain Health Care Hospital, Fountain Projects and Research Office (FOPRO), Eldoret, Kenya, 19 Diabetes Trials Unit, Oxford Centre for Diabetes, Endocrinology and Metabolism, Radcliffe Department of Medicine, University of Oxford, Oxford, United Kingdom, 20 Department of Clinical Tropical Medicine, Faculty of Tropical Medicine, Mahidol University, Bangkok, Thailand, 21 Department of Molecular Tropical Medicine and Genetics, Faculty of Tropical Medicine, Mahidol University, Bangkok, Thailand, 22 Coordination of Allada Ze Toffo Health Zone, Adjian, Benin, 23 National

**Data Availability Statement:** The data underlying the results presented in the study are available from https://github.com/jwatowatson/COPCOV.

**Funding:** This research was funded by the Wellcome Trust [Grant number 221307/Z/20/Z] through the COVID-19 Therapeutics Accelerator.

**Competing interests:** NJW and LvS are members of the PLOS Medicine Editorial Board. The rest of the authors have declared that no competing interests exist.

**Abbreviations:** COVID-19, Coronavirus Disease 2019; CQ, chloroquine; DBS, dried blood spot; DSMB, Data Safety and Monitoring Board; HCQ, hydroxychloroquine; ITT, intention-to-treat; PP, per protocol; RCT, randomised controlled trial; RR, risk ratio; SAE, serious adverse event; SAP, Statistical Analysis Plan; SARS-CoV-2, Severe Acute Respiratory Syndrome Coronavirus 2; SEAC, Serology Endpoint Assessment Committee.

Public Health Laboratory, Cotonou, Benin, **24** Institut Pierre Richet, Institut National de Santé, Publique, Bouaké, Côte d'Ivoire, **25** Centre Hospitalier Universitaire (CHU) de Bouaké, Bouaké, Côte d'Ivoire, **26** Faculty of Medicine, Universitas Airlangga, Surabaya, Indonesia, **27** Universitas Airlangga Teaching Hospital, Universitas Airlangga, Surabaya, Indonesia, **28** Husada Utama Hospital, Surabaya, Indonesia, **29** Faculty of Medicine, Universitas Sumatra Utara, Medan, Indonesia, **30** Murni Teguh Hospital, Medan, Medan, Indonesia, **31** Bunda Thamrin Hospital, Medan, Indonesia, **32** Department of Internal Medicine, Faculty of Medicine, Public Health And Nursing, Universitas Gadjah Mada/ Dr. Sardjito Hospital, Yogyakarta, Indonesia, **33** Dr. Sardjito Hospital, Yogyakarta, Indonesia, **34** Epicentre, Niamey, Niger, **35** Université Abdou Moumouni de Niamey, Faculté des Science de la Santé, Niamey, Niger, **36** Department of Paediatrics and Child Health, Aga Khan University Hospital, Karachi, Pakistan, **37** Infectious Diseases Department, Oxford University Hospitals NHS Foundation Trust, John Radcliffe Hospital, Oxford, United Kingdom, **38** University of Oxford, Medical Sciences Division, John Radcliffe Hospital, Oxford, United Kingdom, **39** University Hospitals of Coventry and Warwickshire NHS Trust, Coventry, United Kingdom, **40** Clinical Research Department, Faculty of Infectious and Tropical Diseases, London School of Hygiene & Tropical Medicine, London, United Kingdom

¶ Membership of COPCOV Collaborative Group provided in Supporting Information file S1 Text.
* william@tropmedres.ac

# Abstract

## Background

Hydroxychloroquine (HCQ) has proved ineffective in treating patients hospitalised with Coronavirus Disease 2019 (COVID-19), but uncertainty remains over its safety and efficacy in chemoprevention. Previous chemoprevention randomised controlled trials (RCTs) did not individually show benefit of HCQ against COVID-19 and, although meta-analysis did suggest clinical benefit, guidelines recommend against its use.

## Methods and findings

Healthy adult participants from the healthcare setting, and later from the community, were enrolled in 26 centres in 11 countries to a double-blind, placebo-controlled, randomised trial of COVID-19 chemoprevention. HCQ was evaluated in Europe and Africa, and chloroquine (CQ) was evaluated in Asia, (both base equivalent of 155 mg once daily). The primary endpoint was symptomatic COVID-19, confirmed by PCR or seroconversion during the 3-month follow-up period. The secondary and tertiary endpoints were: asymptomatic laboratory-confirmed Severe Acute Respiratory Syndrome Coronavirus 2 (SARS-CoV-2) infection; severity of COVID-19 symptoms; all-cause PCR-confirmed symptomatic acute respiratory illness (including SARS-CoV-2 infection); participant reported number of workdays lost; genetic and baseline biochemical markers associated with symptomatic COVID-19, respiratory illness and disease severity (not reported here); and health economic analyses of HCQ and CQ prophylaxis on costs and quality of life measures (not reported here).

The primary and safety analyses were conducted in the intention-to-treat (ITT) population. Recruitment of 40,000 (20,000 HCQ arm, 20,000 CQ arm) participants was planned but was not possible because of protracted delays resulting from controversies over efficacy and adverse events with HCQ use, vaccine rollout in some countries, and other factors. Between 29 April 2020 and 10 March 2022, 4,652 participants (46% females) were enrolled (HCQ/CQ $n = 2,320$; placebo $n = 2,332$). The median (IQR) age was 29 (23 to 39) years. SARS-CoV-2 infections (symptomatic and asymptomatic) occurred in 1,071 (23%)

participants. For the primary endpoint the incidence of symptomatic COVID-19 was 240/2,320 in the HCQ/CQ versus 284/2,332 in the placebo arms (risk ratio (RR) 0.85 [95% confidence interval, 0.72 to 1.00; $p = 0.05$]).

For the secondary and tertiary outcomes asymptomatic SARS-CoV-2 infections occurred in 11.5% of HCQ/CQ recipients and 12.0% of placebo recipients: RR: 0.96 (95% CI, 0.82 to 1.12; $p = 0.6$). There were no differences in the severity of symptoms between the groups and no severe illnesses. HCQ/CQ chemoprevention was associated with fewer PCR-confirmed all-cause respiratory infections (predominantly SARS-CoV-2): RR 0.61 (95% CI, 0.42 to 0.88; $p = 0.009$) and fewer days lost to work because of illness: 104 days per 1,000 participants over 90 days (95% CI, 12 to 199 days; $p < 0.001$). The prespecified meta-analysis of all published pre-exposure RCTs indicates that HCQ/CQ prophylaxis provided a moderate protective benefit against symptomatic COVID-19: RR 0.80 (95% CI, 0.71 to 0.91). Both drugs were well tolerated with no drug-related serious adverse events (SAEs). Study limitations include the smaller than planned study size, the relatively low number of PCR-confirmed infections, and the lower comparative accuracy of serology endpoints (in particular, the adapted dried blood spot method) compared to the PCR endpoint. The COPCOV trial was registered with ClinicalTrials.gov; number NCT04303507.

## Interpretation

In this large placebo-controlled, double-blind randomised trial, HCQ and CQ were safe and well tolerated in COVID-19 chemoprevention, and there was evidence of moderate protective benefit in a meta-analysis including this trial and similar RCTs.

## Trial registration

ClinicalTrials.gov NCT04303507; ISRCTN Registry ISRCTN10207947.

Author summary

### Why was this study done?

- At the beginning of the Coronavirus Disease 2019 (COVID-19) pandemic, there was an urgent need to find ways to prevent COVID-19.

- Laboratory studies showed that the related 4-aminoquinolines, chloroquine (CQ), and hydroxychloroquine (HCQ), which had been used widely for over 50 years, had antiviral activity against Severe Acute Respiratory Syndrome Coronavirus 2 (SARS-CoV-2).

- HCQ proved ineffective in the treatment of hospitalised patients, and individual RCTs testing COVID-19 prophylaxis did not show benefit of HCQ. However, a meta-analysis of trial data did suggest some efficacy in preventing COVID-19.

- Current evidence-based guidelines using data from the same studies recommend strongly against the use of HCQ for prophylaxis.

- This study aimed to provide a definitive answer whether or not pre-exposure use of these drugs could prevent COVID-19.

**What did the researchers do and find?**

- The COPCOV study was a double-blind placebo-controlled evaluation of CQ and HCQ COVID-19 chemoprevention. It was the largest pre-exposure prophylaxis study in COVID-19.

- We found that CQ and HCQ were well tolerated and safe in prophylaxis. There was some evidence for protection against symptomatic COVID-19, and a reduction in workdays lost to illness.

- Our updated meta-analysis of all chemoprevention studies in COVID-19 confirms that chemoprophylaxis with CQ or HCQ is well tolerated, safe, and provides a moderate beneficial effect in preventing COVID-19.

**What do these findings mean?**

- Although CQ or HCQ are unlikely to be used in COVID-19 prevention at this stage, they could have been deployed with benefit earlier, and they might have value in future pandemics.

- Randomised controlled trial (RCT) evidence is essential in evaluating therapeutics in the context of a pandemic.

- Trials should be facilitated and protected so that evidence is generated rapidly and evidence-based policies can be implemented without delay to allow timely interventions.

## Introduction

In the 4 years since the start of the Coronavirus Disease 2019 (COVID-19) pandemic, the majority of the world's population has been infected. It is estimated conservatively that over 6.9 million people have died from COVID-19 [1]. At the beginning of 2020, there were no vaccines and no specific treatments, and there was substantial global concern about the projected consequences of the developing pandemic. Many existing medicines were proposed as potential therapeutics ("repurposing"). Prominent among these were the 4-aminoquinolines, chloroquine (CQ), and hydroxychloroquine (HCQ), as they had been used widely for decades in the prevention and treatment of malaria and rheumatological conditions, and they had in vitro activity against both SARS-CoV-1 and SARS-CoV-2 (Severe Acute Respiratory Syndrome Coronavirus 2) [2,3]. After initial claims of benefit, their use rapidly became politicised and controversial. This unhelpful milieu was worsened in May 2020 by a prominent false claim of lethal cardiovascular toxicity [4]. The clinical trials evaluating the preventive and curative efficacy of the 4-aminoquinolines were caught, and in many cases damaged, by the controversies and regulatory decisions. Soon afterwards, large randomised controlled trials (RCTs) in patients hospitalised with COVID-19 showed definitively that HCQ treatment did not reduce mortality [5,6]. Although it has become clear that antivirals are most effective early in COVID-19, when viral burdens are highest, whereas anti-inflammatory drugs are beneficial in late disease (hospitalised patients), the negative results from the large RCTs in severe COVID-19 were extrapolated to indicate a lack of efficacy for HCQ in all stages of COVID-19 infection [7,8]. Nevertheless, HCQ was recommended widely as COVID-19 chemoprevention [9]. For

the chemoprevention clinical trials attempting to provide definitive evidence the widely publicised controversies and negative regulatory responses adversely affected recruitment and study conduct. Despite this, some investigators did successfully complete their RCTs [10–23]. Taken together, these studies point towards moderate preventive efficacy even though individually most were underpowered to demonstrate benefit [24], but the evidence is far from conclusive. In contrast, most authorities recommend against HCQ [7,25]. As a result, there still remains substantial uncertainty regarding the true efficacy of HCQ in COVID-19 prophylaxis. This study's aim was to characterise the efficacy, tolerability, and safety of HCQ/CQ pre-exposure prophylaxis in the prevention of COVID-19.

## Methods

### Study design

COPCOV was a multinational double-blind, randomised, placebo-controlled trial of COVID-19 chemoprevention (Fig 1). Potential investigators in 76 countries across the world were contacted to seek their interest and ability to conduct the study and follow the protocol (Fig A1 in S1 Appendix). Hydroxychloroquine was evaluated in Africa and the United Kingdom (UK), and chloroquine was evaluated in Asia. As the drugs have comparable in vitro activities, mode of action, and pharmacokinetic properties, they were considered equivalent [26,27].

### Participants

Initially, in the rapidly spreading pandemic, the focus was on protecting healthcare workers but, as the study progressed, the inclusion criteria were widened (S1 Appendix). The study recruited unvaccinated healthy, nonpregnant, adults aged between 17 and 70 years who were at risk of COVID-19, agreed to the study procedures, could be followed reliably for up to 5 months, and had ready access to an internet-enabled smartphone. Participants with any underlying disease or contraindication to taking 4-aminoquinolines were excluded.

### Randomisation and masking

The trial statistician (MM) generated 2 separate permuted-block randomisation sequences in blocks of 10 stratified in sets of 400 drug kits (each kit comprised 10 blister packs of 10 tablets containing either drug or identical placebo) packed into boxes of 50 (hydroxychloroquine and placebo: Accord Healthcare, London, UK) and 200 (chloroquine and placebo: Utopian Pharmaceutical Co., Bangkok, Thailand) kits by Piramal Healthcare (UK) and Utopian, respectively, according to company regulations. The assignments were concealed from investigators, research staff, and participants. Kits were in sequence of randomisation and allocation was by opening in sequential order. The Statistical Analysis Plan (SAP) was completed and signed before the database lock and subsequent unblinding (S1 Statistical Analysis Plan).

After education about the study and provision of voluntary written informed consent, participants were randomised to receive either CQ or HCQ (depending on study site) or identical matched placebos (1:1 randomisation). The tablets were film-coated to conceal the taste and prevent unblinding. A loading dose of 10 mg base/kg (four 155 mg tablets for a 60 kg subject) was followed by 155 mg daily (equivalent to 250 mg chloroquine phosphate or 200 mg hydroxychloroquine sulphate) for 3 months.

### Procedures

At the initial visit, participants were examined and baseline screening blood samples were taken. Participants were instructed how to use the mobile 'phone application (ePRO, Axiom

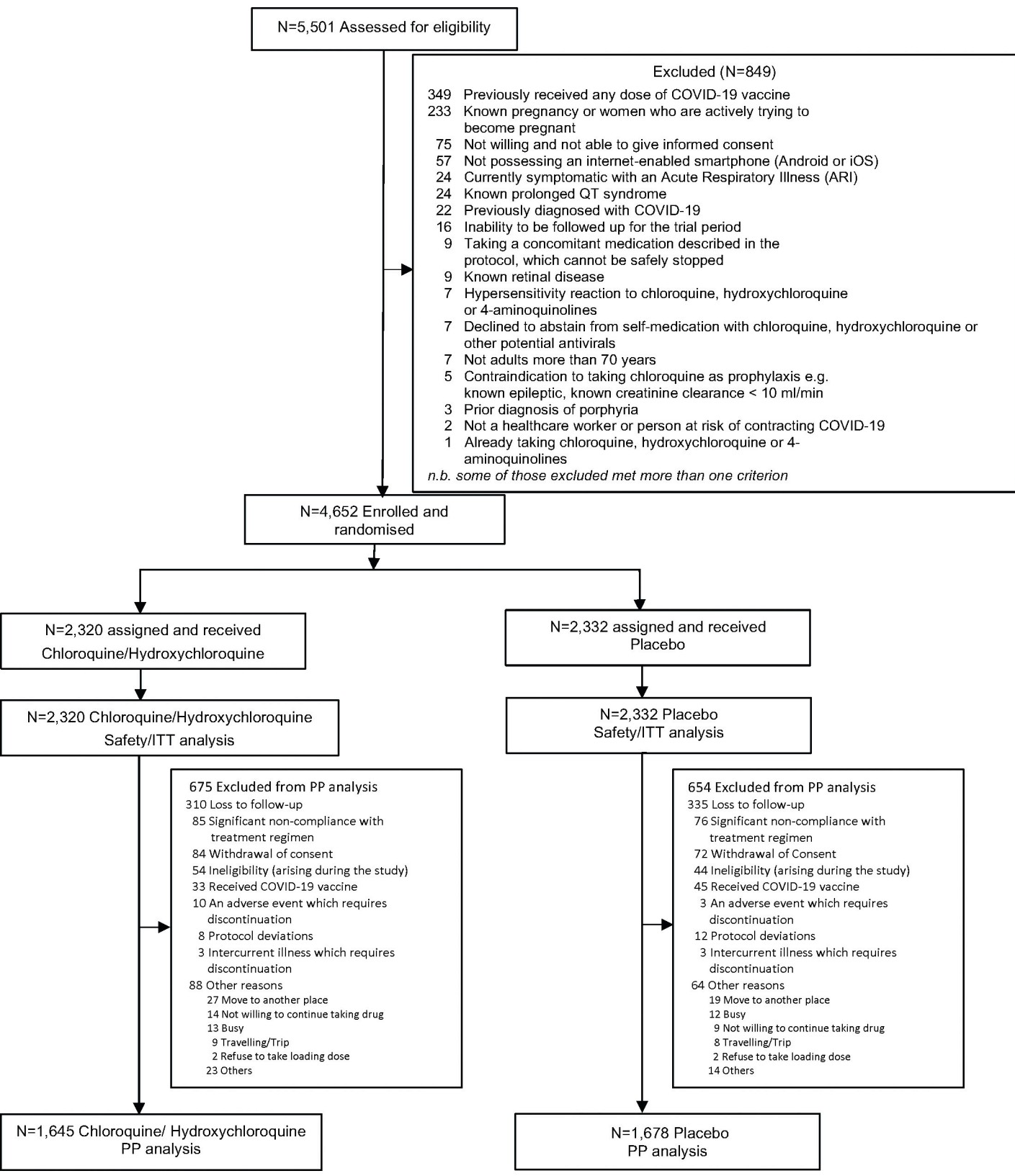

**Fig 1. COPCOV study participant flowchart.**

Real-Time Metrics, ON, Canada) and were asked to record their temperature twice daily (an oral electronic thermometer was provided) and symptoms at least once daily. Thereafter, each participant was reviewed in person each month. Serology serum samples were taken on days 0 and 90, or at the last visit if the participant left the study earlier. Contingency dried blood spot samples (DBS) were taken for drug measurement on days 0, 30, 60, and 90 (the analysis of drug concentrations will be reported separately). If symptoms consistent with COVID-19 occurred, the participant was asked to alert the study team by 'phone, so that nose and throat swabs could be taken. These were stored at −80˚C. The methods for SARS-CoV-2 PCR diagnosis and serological quantification of SARS-CoV-2 spike protein IgG antibodies are described in the S1 Appendix. Vaccines were deployed increasingly during the trial. Vaccinated participants were reviewed and an "end of study" serum sample was taken before, or within 3 days, of their vaccination, and were censored from the trial on receiving the first dose of vaccine.

Procedures for case identification, management, and subsequent isolation followed local and national guidelines. Continuation of the blinded study medication in confirmed COVID-19 cases was at the discretion of the attending health care professional. Overall trial monitoring was conducted by the Mahidol Oxford Tropical Medicine Research Unit Clinical Trials Support Group.

## Outcomes

The primary outcome was symptomatic laboratory-confirmed COVID-19 defined as symptoms consistent with COVID-19 and laboratory evidence of SARS-CoV-2 infection (S1 Appendix). Laboratory evidence was defined by a prespecified hierarchy: first by a nose and/or throat swab PCR positive for SARS-CoV-2; second, if the PCR was negative, failed, or not done, by seroconversion (S-protein IgG antibody) based on baseline and end of study paired sera [28]; and third, only if paired sera were unavailable or uninterpretable, by seroconversion from the contingency DBS in an adapted assay. The prespecified primary endpoint determination algorithm code is provided in the GitHub repository.

A Serology Endpoint Assessment Committee (SEAC) comprising 2 external experts with extensive SARS-CoV-2 serology experience was convened to adjudicate on equivocal serology endpoints (SAP: S1 Appendix). Their judgements before trial unblinding were included in the database and regarded as final.

## Secondary outcomes

There were 3 prespecified secondary outcomes: asymptomatic laboratory-confirmed SARS-CoV-2 infection; severity of COVID-19 symptoms; and all-cause PCR-confirmed symptomatic acute respiratory illness (including SARS-CoV-2 infection).

## Tertiary outcomes

There were 3 generic prespecified trial tertiary outcomes: the participant reported number of workdays lost; genetic and baseline biochemical markers associated with symptomatic COVID-19, respiratory illness, and disease severity (not reported here); and health economic analyses of HCQ and CQ prophylaxis on costs and quality of life measures (not reported here).

In addition, we prespecified that a meta-analysis of previously published randomised hydroxychloroquine COVID-19 pre-exposure chemoprevention studies and the current study should be conducted (S1 Appendix and Methods A2 in S1 Appendix).

## Statistical analysis

At the beginning of the pandemic, we did not know what would be the subsequent incidence of COVID-19, and estimated conservatively a 90-day incidence of 3%. We therefore planned to enrol 20,000 participants in Asia (chloroquine-based randomisation) and 20,000 in Europe/ Africa (hydroxychloroquine-based randomisation). This sample size allowed for approximately 20% loss to follow-up, withdrawals, protocol deviations and non-adherence, and provided 80% power to detect a 23% reduction in symptomatic COVID-19 incidence for each drug individually with 95% confidence. Unfortunately, there were protracted delays and difficulties with recruitment as the study was starting in 2020. These were related to adverse publicity and withdrawal of regulatory approvals resulting from falsified claims of frequent serious cardiotoxicity [4,29]. The Data Safety and Monitoring Board (DSMB) then acknowledged that the original sample size would no longer be achievable, but recommended study continuation, with pooling of the HCQ and CQ results. The primary outcome was subsequently changed to include seroconversion. This resulted in a 4-fold higher event rate than initially forecasted (approximately 12%). Assuming a continued 12% event rate in the control arm a total sample size of 4,600 had >80% power to detect the previously targeted treatment effect (i.e., 23% reduction).

The primary outcome included all participants and was analysed using intention-to-treat (ITT) with a two-sided *p*-value <0.05 considered significant. A secondary per protocol (PP) analysis excluded participants as described in the CONSORT diagram (Fig 1) and in the SAP (S1 Appendix). Secondary and tertiary endpoints were analysed using the ITT population. Fisher's exact test was used to compare treatment effects between groups. Risk ratios were obtained from a log-binomial model. Kaplan–Meier survival curves were estimated for the time to PCR-confirmed symptomatic COVID-19 and all-cause respiratory infection and were tested using the log-rank test. The number of workdays lost was analysed under a zero-inflated Poisson regression model. For convergence reasons, we fitted this model using a Bayesian framework in the *brms* R package (which uses stan to estimate posterior distributions under weakly informative priors). Data analysis was performed using Stata 17.0, StataCorp, College Station, Texas, United States of America and R version 4.2.2. All code and data required to reproduce the primary analysis are provided in https://github.com/jwatowatson/COPCOV. The study is reported according to the CONSORT guidelines for reporting of a randomised trial (S1 CONSORT Checklist).

## Trial support and role of the funding source

The COPCOV trial was approved by all local ethics committees and the Oxford Tropical Research Ethics Committee (OxTREC: 25–20) and was sponsored by the University of Oxford. The DSMB met before and during the trial and reviewed all serious adverse events (SAEs). The study sponsor, funder, and the drug manufacturers had no input into the study design, conduct, oversight, analysis, or reporting. The authors vouch for the completeness and accuracy of the data. After critical review all authors agreed to submit the manuscript for publication.

## Results

The COPCOV study was conducted in 26 sites in 11 countries (Benin, Côte d'Ivoire, Indonesia, Kenya, Mali, Nepal, Niger, Pakistan, Thailand, UK, and Zambia) (Table A1 and Fig A2 In S1 Appendix) and ran between 29 April 2020 and 10 March 2022 (Fig A3 in S1 Appendix). A total of 4,652 adult participants (Table 1) were randomised to hydroxychloroquine or chloroquine (HCQ: 1,299, CQ: 1,021: total: 2,320) or corresponding matched placebos (N = 2,332) (Fig 1). The median age of the participants was 29 years (interquartile range, 23 to 39). The

**Table 1. Demographic details of the COPCOV study participants.**

| | Chloroquine/hydroxychloroquine | Placebo | Total |
|---|---|---|---|
| | N = 2,320 | N = 2,332 | <u>N</u> = 4,652 |
| Age (years), median (IQR) | 29 (23–39) | 29 (24–39) | 29 (23–39) |
| Sex, n (%) | | | |
| Male | 1,252 (54.0) | 1,267 (54.3) | 2,519 (54.1) |
| Female | 1,067 (46.0) | 1,064 (45.6) | 2,131 (45.8) |
| Not specified | 1 (0) | 1 (0) | 2 (0) |
| Temperature (˚C), mean (SD) | 36.4 (0.4) | 36.4 (0.5) | 36.4 (0.5) |
| Weight (kg), mean (SD) | 64.8 (14.2) | 65.3 (14.7) | 65.1 (14.4) |
| Height (cm), mean (SD) | 164 (9) | 164 (10) | 164 (9) |
| BMI (kg/m$^2$), median (IQR) | 23.1 (20.3–26.9) | 23.3 (20.5–26.9) | 23.2 (20.4–26.9) |
| Smoking, n (%) | | | |
| Yes | 374 (16.12) | 395 (16.94) | 769 (16.53) |
| Never smoked | 1,789 (77.11) | 1,772 (75.99) | 3,561 (76.55) |
| Former smoker | 157 (6.77) | 165 (7.08) | 322 (6.92) |
| COVID-19 in household, n/N (%) | 263 (11.3) | 273 (11.7) | 536 (11.5) |
| **Existing comorbidities** | | | |
| Chronic pulmonary disease (not asthma), n/N (%) | 5 (0.2) | 0 (0) | 5 (0.1) |
| Asthma (physician diagnosed), n/N (%) | 22 (0.9) | 18 (0.8) | 40 (0.9) |
| Chronic kidney disease, n/N (%) | 0 (0) | 0 (0) | 0 (0) |
| Liver disease, n/N (%) | 1 (0.04) | 1 (0.04) | 2 (0.04) |
| AIDS/HIV, n/N (%) | 37 (1.6) | 34 (1.5) | 71 (1.5) |
| Diabetes, n/N (%) | 21 (0.9) | 15 (0.6) | 36 (0.8) |
| Hypertension, n/N (%) | 43 (1.9) | 41 (1.8) | 84 (1.8) |
| Cancer, n/N (%) | 1 (0.04) | 1 (0.04) | 2 (0.04) |
| Condition requiring immunosuppressive drugs, n/N (%) | 1 (0.04) | 0 (0) | 1 (0.02) |
| Ischaemic heart disease, n/N (%) | 2 (0.09) | 0 (0) | 2 (0.04) |
| High cholesterol, n/N (%) | 5 (0.2) | 7 (0.3) | 12 (0.3) |
| Any chronic condition, n (%) | 118 (5.1) | 107 (4.6) | 225 (4.8) |
| **Baseline symptoms** | | | |
| Fever, n (%) | 0 (0) | 0 (0) | 0 (0) |
| Cough, n (%) | 5 (0.2) | 3 (0.1) | 8 (0.2) |
| Sore throat, n (%) | 0 (0) | 1 (0.04) | 1 (0.02) |
| Runny nose (Rhinorrhoea), n (%) | 0 (0) | 1 (0.04) | 1 (0.02) |
| Wheezing, n (%) | 1 (0.04) | 0 (0) | 1 (0.02) |
| Anosmia, n (%) | 0 (0) | 0 (0) | 0 (0) |
| Chest pain, n (%) | 0 (0) | 1 (0.04) | 1 (0.02) |
| Muscle pain (myalgia), n (%) | 2 (0.09) | 5 (0.2) | 7 (0.2) |
| Joint pain (Arthralgia), n (%) | 4 (0.2) | 4 (0.2) | 8 (0.2) |
| Shortness of breath on exertion, n (%) | 0 (0) | 3 (0.13) | 3 (0.06) |
| Shortness of breath at rest, n (%) | 0 (0) | 0 (0) | 0 (0) |
| Fatigue/malaise, n (%) | 0 (0) | 3 (0.13) | 3 (0.06) |
| Itching, n (%) | 0 (0) | 1 (0.04) | 1 (0.02) |
| Headache, n (%) | 1 (0.04) | 5 (0.2) | 6 (0.1) |
| Dizziness, n (%) | 2 (0.09) | 3 (0.1) | 5 (0.1) |
| Visual disturbance, n (%) | 0 (0) | 1 (0.04) | 1 (0.02) |
| Abdominal pain, n (%) | 0 (0) | 2 (0.09) | 2 (0.04) |
| Poor appetite, n (%) | 1 (0.04) | 1 (0.04) | 2 (0.04) |

*(Continued)*

**Table 1.** (Continued)

| | Chloroquine/hydroxychloroquine | Placebo | Total |
|---|---|---|---|
| | *N* = 2,320 | *N* = 2,332 | <u>*N*</u> = 4,652 |
| Nausea, *n* (%) | 0 (0) | 2 (0.09) | 2 (0.04) |
| Vomiting, *n* (%) | 0 (0) | 1 (0.04) | 1 (0.02) |
| Diarrhoea, *n* (%) | 1 (0.04) | 0 (0) | 1 (0.02) |
| Skin rash, *n* (%) | 0 (0) | 2 (0.09) | 2 (0.04) |

population was generally healthy; 4.8% (225/4,652) reported having a chronic disease. There were no significant differences in tolerability, safety, or efficacy between chloroquine and hydroxychloroquine.

## Primary outcome

Symptomatic, laboratory-confirmed COVID-19 during the 3-month follow-up period occurred in 524 participants (11.3%); 240/2,320 (10.3% [95% CI, 9.2 to 11.7]) received HCQ/CQ and 284/2,332 (12.3% [95% CI, 10.9 to 13.6]) received placebo (risk ratio (RR) 0.85 [95% CI, 0.72 to 1.00; *p* = 0.05]) (Table 2). Symptomatic COVID-19 was PCR-confirmed in 24/2,320 HCQ/CQ recipients (1.0% [95%CI, 0.7 to 1.5]) and 56/2,332 (2.4% [95% CI, 1.8 to 3.1]) placebo recipients (RR 0.43 [95% CI, 0.27 to 0.69; *p* < 0.001]) (Fig 3). Among those with paired sera, symptomatic COVID-19 was diagnosed by seroconversion in 211/1,462 (14.4% [95% CI, 12.7 to 16.3]) HCQ/CQ recipients and 245/1,498 (16.4% [95%CI, 14.5 to 18.3]) placebo recipients (RR 0.88 [95% CI, 0.74 to 1.05], *p* = 0.2) (Table 2). In the remaining cases analysed by DBS (i.e., without paired sera), symptomatic COVID-19 was diagnosed in 25/280 (8.9% [95% CI, 5.9 to 12.9]) HCQ/CQ recipients and 26/297 (8.8 [95% CI, 5.8 to 12.6]) placebo recipients (RR 1.02 [95% CI, 0.60 to 1.72; *p* = 0.9]). Primary outcome data were missing for 1,114 participants (24%) for the serology component, mainly because of loss to follow-up (Fig 2). Asymptomatic infection was also common, occurring in 524 participants, thus the overall proportion of SARS-CoV-2 infection within the 3-month study was 23% (1,071 of 4,652).

## Prespecified secondary outcomes

Asymptomatic COVID-19 occurred in 267/2,320 (11.5% [95% CI, 10.2 to 12.9]), HCQ/CQ recipients and 280/2,332 (12.0% [95% CI, 10.7 to 13.4]) placebo recipients (RR: 0.96 [95% CI, 0.82 to 1.12; *p* = 0.6]). In symptomatic patients, there were no significant differences in symptom severity scores (20.0 (5–85) versus 21.5 (5–89)). All-cause respiratory illness (mainly COVID-19) occurred in 44/2,320 (1.9% [95% CI, 1.4 to 2.5]) HCQ/CQ recipients and 73/2,332 (3.1% [95% CI, 2.5 to 3.9]) placebo recipients (RR: 0.61 [95% CI, 0.42 to 0.88; *p* = 0.009]) (Fig 3 and Table A8 in S1 Appendix).

## Prespecified tertiary outcomes

Under a zero-inflated Poisson regression model, the mean number of workdays lost over 90 days of chemoprophylaxis was 337 (95% CI, 279 to 398) per 1,000 participants in HCQ/CQ recipients and 441 (95% CI, 370 to 515) per 1,000 in placebo recipients; a mean difference of 104 days (95% CI, 12 to 199). Similar results were observed in the ITT and per protocol analyses for all comparisons (Tables A2–A3 in S1 Appendix). Tests for genetic and biochemical markers for symptomatic COVID-19, respiratory illness or disease severity, were not

**Table 2. Prespecified endpoints of the COPCOV trial in the intention to treat population.**

| Outcome | Chloroquine/ hydroxychloroquine (N = 2,320) | Placebo (N = 2,332) | Risk ratio (95% CI) | Fisher's exact P-value |
|---|---|---|---|---|
| Total participant days | 181,263 | 184,688 | | |
| **Primary endpoint:** Symptomatic laboratory-confirmed COVID-19. n (%); 95% CI | 240/2,320 10.3 (9.1 to 11.7) | 284/2,332 12.2 (10.9 to 13.6) | 0.85 (0.72 to 1.00) | 0.051 |
| PCR-confirmed diagnosis. n/N (%); 95% CI | 24/2,320 1.0 (0.7 to 1.5) | 56/2,332 2.4 (1.8 to 3.1) | 0.43 (0.27 to 0.69) | <0.001 |
| Serology confirmed diagnosis (serum). n (%); 95% CI | 211/1,462 14.4 (12.7 to 16.3) | 245/1,498 16.4 (14.5 to 18.3) | 0.88 (0.74 to 1.05) | 0.154 |
| Serology confirmed diagnosis (DBS). n (%); 95% CI | 25/280 8.9 (5.9–12.9) | 26/297 8.8 (5.8 to 12.6) | 1.02 (0.60 to 1.72) | 1.000 |
| **Secondary endpoints:** | | | | |
| Asymptomatic SARS-CoV-2 infection. n (%); 95% CI | 267/2,320 11.5 (10.2 to 12.9) | 280/2,332 12.0 (10.7 to 13.4) | 0.96 (0.82 to 1.12) | 0.617 |
| All SARS-CoV-2 infection n (%); 95% CI | 507/2,320 21.9 (20.2 to 23.6) | 564/2,332 24.2 (22.5 to 26.0) | 0.90 (0.81 to 1.00) | 0.060 |
| All-cause respiratory illness*. n (%); 95% CI | 44/2,320 1.9 (1.4 to 2.5) | 73/2,332 3.1 (2.5 to 3.9) | 0.61 (0.42 to 0.88) | 0.009 |
| Severity score. Median (IQR) | 20.0 (5–85) | 21.5 (5–89) | NA | 1.000 |
| **Tertiary endpoint:** Participant reported workdays lost | 700/181,263 | 932/184,688 | NA | 0.0002** |

*PCR-confirmed respiratory infection including COVID-19.

**Assessed by a zero-inflated Poisson regression model.

Missing outcomes in the primary endpoint ITT analysis were treated as not having had COVID-19 during the study period.

See Fig 2 for Venn diagram depicting breakdown of numbers analysed.

conducted, as suitable specific tests were not available. A health economic analysis will be reported separately.

## Safety and tolerability

The study medications were well tolerated. One HCQ recipient was hospitalised with confirmed COVID-19. There were 10 SAEs in 9 HCQ/CQ recipients and 8 SAEs in 8 placebo recipients but none were considered drug-related (Table 3 and Table A4 in S1 Appendix) and no one died. Overall, 218/2,320 (9.4%) HCQ/CQ recipients had at least 1 adverse event (AE) compared to 242/2,332 (10.4%) placebo recipients (p = 0.3). Fewer HCQ/CQ recipients had severe AEs (31/2,320; 1.3%) than placebo recipients (58/2,332; 2.5%; p = 0.005). Twelve HCQ/CQ recipients (0.52%) discontinued chemoprophylaxis either because of side-effects (N = 10) or inability to comply with study procedures (N = 2), versus 4 (0.17%) in the placebo group (p = 0.05).

## Prespecified meta-analysis

Including this study and the 11 previously published randomised hydroxychloroquine COVID-19 pre-exposure chemoprevention studies (Fig 3), the prespecified meta-analysis

Total participants: *n*=4652

**Fig 2. Venn diagram depicting the breakdown of numbers analysed.** Those with no serology data were either those who dropped out of the study before day 30, and had no end of study samples, or had missing end of study samples, or those judged unreliable by the serology endpoint assessment committee. This group differed from the PP population, with some overlap.

**Table 3. Safety and tolerability of the COPCOV study medications.**

| Adverse events | Chloroquine/ hydroxychloroquine | | Placebo | | Fisher's exact *P*-value |
|---|---|---|---|---|---|
| Number of subjects | *N* = 2,320 | | *N* = 2,332 | | |
| Total participant days | 181,263 | | 184,688 | | |
| Number of subjects with at least 1 AE, *n* (%) <br> Total adverse events, $n_E$ (%) | 218 (9.4) <br> 578 (24.9) | | 242 (10.4) <br> 656 (28.1) | | 0.260 |
| Participants with at least 1 SAE, *n* (%): <br> Total events, $n_E$ (%): | 9 (0.4) <br> 10 (0.4) | | 8 (0.3) <br> 8 (0.3) | | |
| Deaths, *n/N*, (%) | 0 (0) | | 0 (0) | | |
| Possible, probable, or definite drug related SAEs, *n/N*, (%) | 0 (0) | | 0 (0) | | |
| **Grading of adverse events, $n_E$/N, (%)** | **Moderate (grade 2) N = 2,320** | **Severe (grade 3) N = 2,320** | **Moderate (grade 2) N = 2,332** | **Severe (grade 3) N = 2,332** | **P-value for total severe AEs between groups** |
| Symptoms | | | | | |
| Number of adverse events** | 547 (23.6) | 31 (1.3) | 598 (25.6) | 58 (2.5) | 0.005 |
| Itching | 7 (0.3) | 0 (0) | 11 (0.5) | 1 (0) | |
| Headache | 85 (3.7) | 5 (0.2) | 85 (3.6) | 11 (0.5) | |
| Dizziness | 30 (1.3) | 1 (0) | 20 (0.9) | 0 (0) | |
| Visual disturbance | 3 (0.1) | 0 (0) | 7 (0.3) | 0 (0) | |
| Abdominal pain | 22 (0.9) | 5 (0.2) | 19 (0.8) | 6 (0.3) | |
| Poor appetite | 3 (0.1) | 0 (0) | 5 (0.2) | 0 (0) | |
| Nausea | 19 (0.8) | 0 (0) | 19 (0.8) | 2 (0.1) | |
| Vomiting | 9 (0.4) | 1 (0) | 8 (0.3) | 1 (0) | |
| Diarrhoea | 22 (0.9) | 0 (0) | 17 (0.7) | 2 (0.1) | |
| Skin rash | 6 (0.3) | 0 (0) | 3 (0.1) | 1 (0) | |
| Other | 341 (14.7) | 19 (0.8) | 404 (17.3) | 34 (1.5) | |

*There were no participants with grade 4 AEs.

**SAEs are reported separately in the same table.

SAE, serious adverse event.

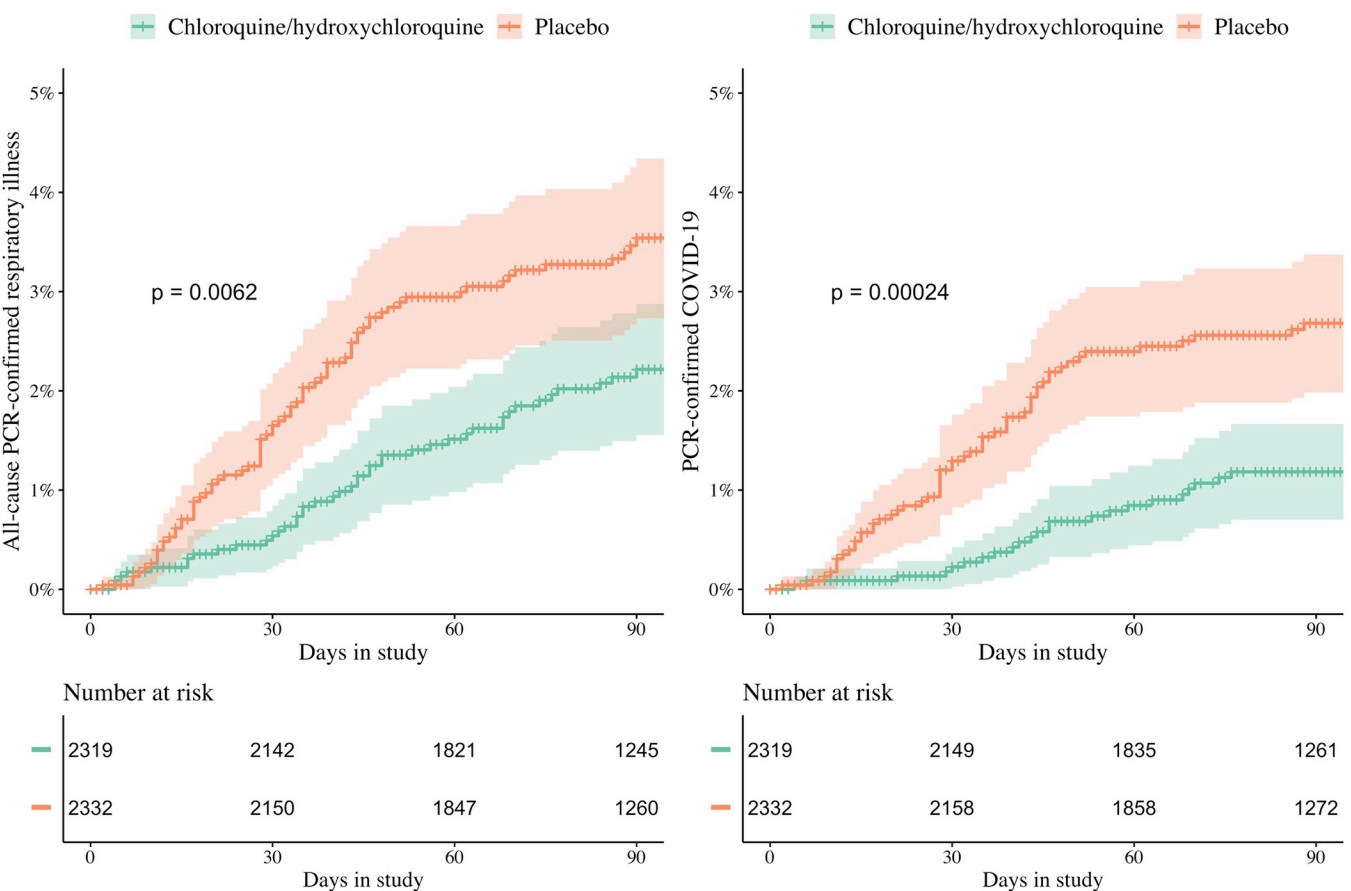

**Fig 3.** All-cause PCR-confirmed respiratory infections (left) and PCR-confirmed COVID-19 (right) over time in the HCQ/CQ recipients (green) and placebo recipients (pink). All-cause respiratory infection was a secondary endpoint. The majority (68%, 80/117) of infections were SARS-CoV-2. Log-rank test p-values are shown. Patients are right censored at the date of last visit. COVID-19, Coronavirus Disease 2019; CQ, chloroquine; HCQ, hydroxychloroquine; SARS-CoV-2, Severe Acute Respiratory Syndrome Coronavirus 2.

showed an overall protective efficacy against symptomatic COVID-19 (RR 0.80 [95% CI, 0.71 to 0.91; $p = < 0.001$]) [11,12,15–23]. Although the studies reported slightly different outcomes, used different doses and for different durations, there was no clear between-study heterogeneity or evidence of publication bias (Fig A4 in S1 Appendix).

## Discussion

This large multinational randomised double-blind COVID-19 chemoprevention trial provides strong evidence of the safety of hydroxychloroquine and chloroquine given daily for 3 months (average 2.4 mg base/kg/day). The trial ended below its original recruitment objective but, because of the higher than anticipated incidence of COVID-19, it was able to provide evidence relevant to protective efficacy. For the trial's primary endpoint (incidence of symptomatic laboratory-confirmed COVID-19), the apparent benefit observed for HCQ/CQ treatment is consistent with the aggregated results of previous smaller studies [11,12,15–23], all but one of which had a lower incidence of symptomatic COVID-19 in the HCQ treatment arm (Fig 4). Incorporating the COPCOV trial data in a prespecified meta-analysis of pre-exposure RCTs, in which there was little heterogeneity across the studies and no evidence of publication bias, suggests a moderate protective benefit (RR 0.80 [95% CI 0.71 to 0.91]). In other prespecified

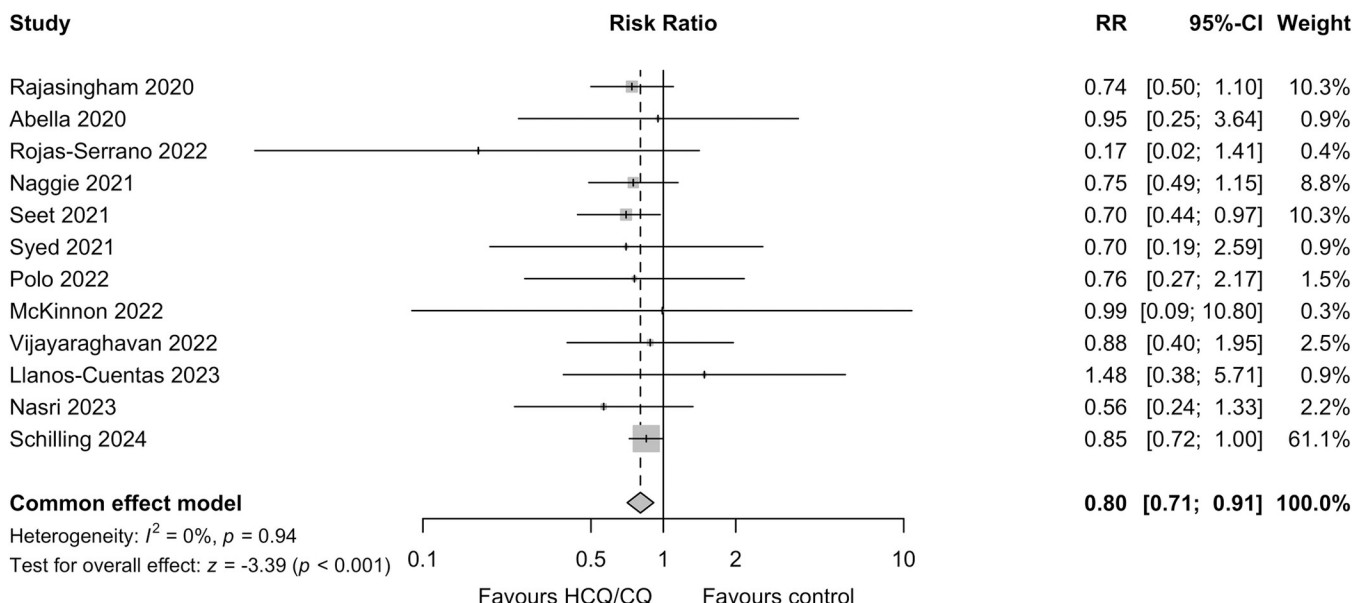

**Fig 4. Prespecified meta-analysis of 4-aminoquinoline COVID-19 pre-exposure chemoprevention RCTs based on individual study primary endpoints according to the method of Garcia-Albeniz and colleagues [24].** Schilling 2024 refers to the current study. The size of the grey squares centred at the treatment effect estimates are proportional to the study weight. Risk ratios were determined for all studies based on the reported data, apart from Seet and colleagues, which was cluster randomised and a recalculated adjusted RR was used. See Methods A2 in S1 Appendix for further details. CQ, chloroquine; HCQ, hydroxychloroquine; RCT, randomised controlled trial; RR, risk ratio.

comparisons in the COPCOV study, HCQ/CQ chemoprevention was associated with fewer adverse events, fewer all-cause PCR-confirmed respiratory infections (mostly COVID-19), and a reduction in workdays lost, but it did not affect the incidence of asymptomatic infections or the severity of respiratory illness (although no patients required hospitalisation for hypoxia).

Despite this study's size, and the combined evidence, there is still substantial uncertainty as to the true prophylactic benefit of these 4-aminoquinolines in COVID-19. The large difference between the groups in rates of PCR-confirmed diagnosis was not mirrored by a similar difference in seroconversion-based diagnosis. This could be a chance finding from PCR (as numbers were relatively small) or, more likely, results from imprecision of serodiagnosis diluting the power of the study. The DBS samples, used to determine seroconversion when paired sera were unavailable, may have provided unreliable results. The extraction of serum from DBS is variable and inefficient and, unlike the serum-based assays [28], the seroconversion thresholds have not been fully validated, although they have been used in other COVID-19 studies [30]. The absence of a significant protective effect against asymptomatic infection may indicate that protective benefit is proportional to the viral burden and thus disease severity, but it could also reflect the imprecision of the seroconversion-based endpoint.

The tolerability and safety of these well-established 4-aminoquinoline medicines is further reinforced in this double-blind comparison [31]. This was a point of controversy early in the pandemic. Initial reports described a range of toxicities that created a climate of concern and adverse opinion. This trial provides no support for these claims, and it supports the generally good short-term safety profile of these drugs [31]. It challenges the World Health Organization "living" guidelines on COVID-19 chemoprevention which emphasised toxicity in recommending against hydroxychloroquine. The same guideline also stated that prophylactic use "probably has a small or no effect on laboratory-confirmed COVID-19" with "moderate

certainty," and has no effect on mortality with "high certainty" [27]. Yet, the meta-analysis reported here suggests significant benefit. No deaths have been reported in any of the pre-exposure prophylaxis studies, and the drugs have been well tolerated with very low rates of treatment discontinuation (Figs A5–A6 in S1 Appendix) [32].

Few drugs have excited such controversy as hydroxychloroquine in COVID-19. Bureaucracy, politicisation, and polarised debate obstructed the undertaking of clinical trials needed to provide objective evidence early in the pandemic. Now, with increasing vaccine coverage, declining viral virulence and availability of effective medicines, there is little reason to recommend the 4-aminoquinolines for COVID-19 chemoprevention. But earlier in the pandemic mortality was much higher, there were no vaccines or drugs, and healthcare systems were under intense strain as their workforces were depleted by illness. If these results had been available then, a case could have been made to deploy these moderately effective, inexpensive, available, and safe 4-aminoquinolines more widely. To put the moderate prophylactic efficacy of HCQ/CQ in perspective, the WHO stated initially that COVID-19 vaccines with a protective efficacy of at least 30% would be considered [33]. In those early days, the rapid spread and the initial high mortality of COVID-19 in the absence of proven effective therapies or vaccines created global concern but, despite the seriousness of the pandemic, there were major obstacles to the conduct of urgently needed clinical research, particularly in low resource settings. In addition, the endorsement of unproven medicines and remedies (including hydroxychloroquine), and later warnings and revocations by regulatory authorities [29], the associated politicisation, and the intense media scrutiny excited controversy and polarised views. All of these factors compromised objective scientific evaluation.

There are significant limitations to this study. The final sample size was substantially smaller than intended which limits confidence in the preventive effect estimates. At the beginning of the pandemic we did not know what the incidence of COVID-19 would be, nor what magnitude of preventive benefit would be considered enough to warrant use of these medicines in prophylaxis. The trial was conducted in many resource-limited settings where collection of respiratory swab specimens and PCR diagnosis was difficult, so only a small proportion of cases could be confirmed by PCR. Seroconversion over 3 months is a less precise measure of symptomatic infection. In particular, the DBS method of assessment, although employed in other studies [30], was not validated and did not perform well in this study. A significant proportion of participants were lost to follow-up (Fig 1) although these were balanced across arms (Fig A7 in S1 Appendix) and the PP analysis corresponded well with that of the ITT population (Tables A2–A3 in S1 Appendix). In addition, some participant outcomes were not evaluable because the end-of-study results were either unavailable (early drop-out), or were considered to be unreliable by the independent Serology Endpoint Assessment Committee (Table 2 and Table A5 in S1 Appendix). Although the inclusion of the DBS assessment was prespecified in the composite endpoint, this likely reduced the precision of the endpoint ascertainment and thus the estimate of preventive efficacy (Table 2). The study primary endpoints evaluated in the meta-analysis were slightly different, but meta-analysis of the PCR-confirmed infections indicates consistent evidence of moderate protective benefit (Fig A6 in S1 Appendix).

In summary, this large double-blind, placebo-controlled, trial showed that hydroxychloroquine and chloroquine prophylaxis was safe and well tolerated, and combined with data from other similar trials provided some evidence that laboratory-confirmed symptomatic COVID-19 might be reduced. The totality of evidence from this and other RCTs suggests a moderate preventive benefit. This knowledge could have proved beneficial earlier in the COVID-19 pandemic when vaccines and treatments were lacking, and mortality and morbidity were high. RCTs should be supported during pandemics.

## Supporting information

**S1 CONSERVE Checklists. CONSERVE Checklists.**
(DOCX)

**S1 CONSORT Checklist. CONSORT 2010 checklist of information to include when reporting a randomised trial.**
(DOCX)

**S1 Protocol. Chloroquine/hydroxychloroquine prevention of coronavirus disease (COVID-19) in the healthcare setting; a randomised, placebo-controlled prophylaxis study (COPCOV).**
(DOCX)

**S1 RiskOfBias Assessment. Risk of Bias Assessment.**
(XLSX)

**S1 Statistical Analysis Plan. Statistical Analysis Plan.**
(PDF)

**S1 Text. Membership of COPCOV Collaborative Group.**
(XLSX)

**S1 Appendix. Table A1.** List of COPCOV study sites. Table A2. Baseline characteristics in the COPCOV trial (Per Protocol Analysis). Table A3. Outcomes of Chloroquine/Hydroxychloroquine and Placebo Pre-exposure Prophylaxis against COVID-19 in the COPCOV study (Per Protocol Analysis). Table A4. Summary of Serious Adverse Events in the COPCOV study. Table A5. Primary and secondary outcomes of Chloroquine/Hydroxychloroquine Therapy for Pre-exposure Prophylaxis against COVID-19 (missing outcomes treated as not having had COVID-19 during the study period) ITT–Results presented as "Risk differences." Table A6. Outcomes of Chloroquine/Hydroxychloroquine and Placebo Pre-exposure Prophylaxis against COVID-19 in the COPCOV study (removing cases for which the SEAC judged that a study endpoint could not be determined). Table A7. Summary characteristics of previously published pre-exposure prophylaxis studies considered for meta-analysis. Table A8. Listing of causes of PCR-confirmed respiratory illness. Fig A1. Atlas showing those countries in which investigators were contacted to enquire whether they would be interested in, and able to join the COPCOV study. Fig A2. Atlas showing the location of the COPCOV trial sites which recruited participants, the 4-aminoquinoline tested, and the approximate numbers recruited. Fig A3. Graph showing cumulative enrollment over time (per week) by country. Fig A4. Funnel plot showing the 4-aminoquinoline COVID-19 pre-exposure chemoprevention RCTs included in the prespecified meta-analysis, and the relationship between point estimate risk ratios for the primary outcome and the corresponding standard errors. Fig A5. Meta-analysis of 4-aminoquinoline COVID-19 pre-exposure chemoprevention RCTs based on individual study primary endpoints using Risk Of Bias tool (RoB 2). Fig A6. Meta-analysis of the safety and tolerability outcomes in COVID-19 chemoprevention RCTs using the same methodology as reported in the WHO living guideline [24]. Fig A7. Meta-analysis of adverse events leading to treatment discontinuation reported in double-blind, placebo-controlled, 4-aminoquinoline COVID-19 pre-exposure chemoprevention RCTs. Fig A8. Graph showing cumulative loss to follow-up (LTFU) for Hydroxychloroquine/ Chloroquine arms and Placebo arm.
(DOCX)

## Acknowledgments

We are very grateful to all the participants of this study and the many nurses, technicians, laboratory scientists, doctors, and support staff who helped with the study.

## Author Contributions

**Conceptualization:** William H. K. Schilling, Arjun Chandna, Arjen Dondorp, Nicholas P. J. Day, Nicholas J. White.

**Data curation:** William H. K. Schilling, Mavuto Mukaka, Thatsanun Ngernseng, Naomi Waithira, James A. Watson, Otieno Edwin Onyango, Elizabeth M. Batty.

**Formal analysis:** William H. K. Schilling, Mavuto Mukaka, James A. Watson, Arjun Chandna, Janjira Thaipadungpanit, Panuvit Rienpradub, Mohammad Yazid Abdad, Elizabeth M. Batty, Arjen Dondorp, Nicholas P. J. Day, Nicholas J. White.

**Funding acquisition:** William H. K. Schilling, Weerapong Phumratanaprapin, Arjen Dondorp, Nicholas P. J. Day, Nicholas J. White.

**Investigation:** William H. K. Schilling, James J. Callery, Martin J. Llewelyn, Cintia V. Cruz, Mehul Dhorda, Maneerat Ekkapongpisit, Erni J. Nelwan, Raph L. Hamers, Anthony Etyang, Mohammad Asim Beg, Samba Sow, William Yavo, Aurel Constant Allabi, Buddha Basnyat, Sanjib Kumar Sharma, Modupe Amofa-Sekyi, Paul Yonga, Amanda Adler, Janjira Thaipadungpanit, Panuvit Rienpradub, Mallika Imwong, Mohammad Yazid Abdad, Stuart D. Blacksell, Joel Tarning, Frejus Faustin Goudjo, Ange D. Dossou, Abibatou Konaté-Touré, Serge-Brice Assi, Kra Ouffoué, Nasronudin Nasronudin, Brian Eka Rachman, Pradana Zaky Romadhon, Didi Darmahadi Dewanto, Made Oka Heryana, Theresia Novi, Ayodhia Pitaloka Pasaribu, Mutiara Mutiara, Miranda Putri Rahayu Nasution, Khairunnisa Khairunnisa, Fauzan Azima Dalimunthe, Eka Airlangga, Akmal Fahrezzy, Yanri Subronto, Nur Rahmi Ananda, Mutia Rahardjani, Atika Rimainar, Ruth Khadembu Lucinde, Molline Timbwa, Clara Agutu, Samuel Akech, Mainga Hamaluba, Jairus Kipyego, Obadiah Ngachi, Fadima Cheick Haidara, Oumar Y. Traoré, François Diarra, Basudha Khanal, Piyush Dahal, Suchita Shrestha, Samita Rijal, Youssouf Kabore, Ousmane Guindo, Farah Naz Qamar, Abdul Momin Kazi, Charles J. Woodrow, Steven Laird, Maina Cheeba, Helen Ayles, Kesinee Chotivanich, Sasithon Pukrittayakamee, Weerapong Phumratanaprapin, Nicholas J. White.

**Methodology:** William H. K. Schilling, James J. Callery, Martin J. Llewelyn, Cintia V. Cruz, Mehul Dhorda, Arjun Chandna, Erni J. Nelwan, Raph L. Hamers, Amanda Adler, Tanya Cope, Janjira Thaipadungpanit, Panuvit Rienpradub, Mallika Imwong, Mohammad Yazid Abdad, Stuart D. Blacksell, Joel Tarning, Charles J. Woodrow, Phaik Yeong Cheah, Walter R. J. Taylor, Lorenz von Seidlein, Arjen Dondorp, Nicholas P. J. Day, Nicholas J. White.

**Project administration:** William H. K. Schilling, James J. Callery, Cintia V. Cruz, Mehul Dhorda, Maneerat Ekkapongpisit, Erni J. Nelwan, Anthony Etyang, Samba Sow, William Yavo, Aurel Constant Allabi, Buddha Basnyat, Sanjib Kumar Sharma, Modupe Amofa-Sekyi, Paul Yonga, Amanda Adler, Prayoon Yuentrakul, Tanya Cope, Frejus Faustin Goudjo, Ange D. Dossou, Abibatou Konaté-Touré, Serge-Brice Assi, Kra Ouffoué, Nasronudin Nasronudin, Brian Eka Rachman, Pradana Zaky Romadhon, Didi Darmahadi Dewanto, Made Oka Heryana, Theresia Novi, Ayodhia Pitaloka Pasaribu, Mutiara Mutiara, Miranda Putri Rahayu Nasution, Khairunnisa Khairunnisa, Fauzan Azima Dalimunthe, Eka Airlangga, Akmal Fahrezzy, Yanri Subronto, Nur Rahmi Ananda, Mutia Rahardjani, Atika Rimainar, Ruth Khadembu Lucinde, Molline Timbwa, Otieno Edwin Onyango,

Clara Agutu, Samuel Akech, Mainga Hamaluba, Jairus Kipyego, Fadima Cheick Haidara, Oumar Y. Traoré, François Diarra, Basudha Khanal, Piyush Dahal, Suchita Shrestha, Samita Rijal, Youssouf Kabore, Ousmane Guindo, Farah Naz Qamar, Maina Cheeba, Kesinee Chotivanich, Nicholas J. White.

**Resources:** Thatsanun Ngernseng, Naomi Waithira, Maneerat Ekkapongpisit, Anthony Etyang, Mohammad Asim Beg, Janjira Thaipadungpanit, Mallika Imwong, Mohammad Yazid Abdad, Stuart D. Blacksell, Eric Adehossi, Helen Ayles, Phaik Yeong Cheah, Sasithon Pukrittayakamee, Weerapong Phumratanaprapin, Nicholas P. J. Day, Nicholas J. White.

**Software:** Mavuto Mukaka, Thatsanun Ngernseng, Naomi Waithira, Elizabeth M. Batty.

**Supervision:** William H. K. Schilling, Mavuto Mukaka, James J. Callery, Martin J. Llewelyn, Mehul Dhorda, Thatsanun Ngernseng, Naomi Waithira, Erni J. Nelwan, Raph L. Hamers, Anthony Etyang, Mohammad Asim Beg, Samba Sow, William Yavo, Aurel Constant Allabi, Buddha Basnyat, Sanjib Kumar Sharma, Modupe Amofa-Sekyi, Amanda Adler, Prayoon Yuentrakul, Tanya Cope, Janjira Thaipadungpanit, Mallika Imwong, Mohammad Yazid Abdad, Stuart D. Blacksell, Joel Tarning, Ayodhia Pitaloka Pasaribu, Yanri Subronto, Atika Rimainar, Molline Timbwa, Mainga Hamaluba, Basudha Khanal, Piyush Dahal, Eric Adehossi, Farah Naz Qamar, Abdul Momin Kazi, Charles J. Woodrow, Steven Laird, Helen Ayles, Walter R. J. Taylor, Kesinee Chotivanich, Sasithon Pukrittayakamee, Weerapong Phumratanaprapin, Lorenz von Seidlein, Arjen Dondorp, Nicholas P. J. Day, Nicholas J. White.

**Validation:** William H. K. Schilling, Mavuto Mukaka, Thatsanun Ngernseng, James A. Watson, Prayoon Yuentrakul, Tanya Cope, Mallika Imwong, Otieno Edwin Onyango, Walter R. J. Taylor, Lorenz von Seidlein, Nicholas J. White.

**Visualization:** Mavuto Mukaka, James A. Watson, Phaik Yeong Cheah.

**Writing – original draft:** William H. K. Schilling, Mavuto Mukaka, James J. Callery, Cintia V. Cruz, Mehul Dhorda, James A. Watson, Lorenz von Seidlein, Arjen Dondorp, Nicholas P. J. Day, Nicholas J. White.

**Writing – review & editing:** William H. K. Schilling, Mavuto Mukaka, James J. Callery, Martin J. Llewelyn, Cintia V. Cruz, Mehul Dhorda, Thatsanun Ngernseng, Naomi Waithira, Maneerat Ekkapongpisit, James A. Watson, Arjun Chandna, Erni J. Nelwan, Raph L. Hamers, Anthony Etyang, Mohammad Asim Beg, Samba Sow, William Yavo, Aurel Constant Allabi, Buddha Basnyat, Sanjib Kumar Sharma, Modupe Amofa-Sekyi, Paul Yonga, Amanda Adler, Prayoon Yuentrakul, Tanya Cope, Janjira Thaipadungpanit, Panuvit Rienpradub, Mallika Imwong, Mohammad Yazid Abdad, Stuart D. Blacksell, Joel Tarning, Frejus Faustin Goudjo, Ange D. Dossou, Abibatou Konaté-Touré, Serge-Brice Assi, Kra Ouffoué, Nasronudin Nasronudin, Brian Eka Rachman, Pradana Zaky Romadhon, Didi Darmahadi Dewanto, Made Oka Heryana, Theresia Novi, Ayodhia Pitaloka Pasaribu, Mutiara Mutiara, Miranda Putri Rahayu Nasution, Khairunnisa Khairunnisa, Fauzan Azima Dalimunthe, Eka Airlangga, Akmal Fahrezzy, Yanri Subronto, Nur Rahmi Ananda, Mutia Rahardjani, Atika Rimainar, Ruth Khadembu Lucinde, Molline Timbwa, Otieno Edwin Onyango, Clara Agutu, Samuel Akech, Mainga Hamaluba, Jairus Kipyego, Obadiah Ngachi, Fadima Cheick Haidara, Oumar Y. Traoré, François Diarra, Basudha Khanal, Piyush Dahal, Suchita Shrestha, Samita Rijal, Youssouf Kabore, Eric Adehossi, Ousmane Guindo, Farah Naz Qamar, Abdul Momin Kazi, Charles J. Woodrow, Steven Laird, Maina Cheeba, Helen Ayles, Phaik Yeong Cheah, Walter R. J. Taylor, Elizabeth M. Batty, Kesinee

Chotivanich, Sasithon Pukrittayakamee, Weerapong Phumratanaprapin, Lorenz von Seidlein, Arjen Dondorp, Nicholas P. J. Day, Nicholas J. White.

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
