## [Editor Report · Decision Letter 0]

25 Aug 2023

Dear Dr Schilling, 

Thank you for submitting your manuscript entitled "Hydroxychloroquine or chloroquine prevention of COVID-19 (COPCOV) a double-blind, randomised, placebo-controlled trial" for consideration by PLOS Medicine.

Your manuscript has now been evaluated by the PLOS Medicine editorial staff and I am writing to let you know that we would like to send your submission out for external assessment.

However, we first need you to complete your submission by providing the metadata that are required for full assessment. To this end, please login to Editorial Manager where you will find the paper in the 'Submissions Needing Revisions' folder on your homepage. Please click 'Revise Submission' from the Action Links and complete all additional questions in the submission questionnaire.

Please re-submit your manuscript within two working days, i.e. by Aug 29 2023 11:59PM.

Once your full submission is complete, your paper will undergo a series of checks in preparation for full assessment.

Kind regards,

Richard Turner PhD

Consulting Editor, PLOS Medicine

plosmedicine@plos.org

---

## [Decision Letter · Decision Letter 1]

8 Oct 2023

Dear Dr. Schilling,

Thank you very much for submitting your manuscript "Hydroxychloroquine or chloroquine prevention of COVID-19 (COPCOV) a double-blind, randomised, placebo-controlled trial" (PMEDICINE-D-23-02452R1) for consideration at PLOS Medicine. 

Your paper was discussed among the editors and sent to independent reviewers, including a statistical reviewer. The reviews are appended at the bottom of this email and any accompanying reviewer attachments can be seen via the link below:

[LINK]

In light of these reviews, we will not be able to accept the manuscript for publication in the journal in its current form, but we would like to invite you to submit a revised version that addresses the reviewers' and editors' comments fully. You will recognize that we cannot make a decision about publication until we have seen the revised manuscript and your response, and we expect to seek re-review by one or more of the reviewers. 

We hope to receive your revised manuscript by Oct 30 2023 11:59PM. Please email us (plosmedicine@plos.org) if you have any questions or concerns.

Please let me know if you have any questions, and we look forward to receiving your revised manuscript. 

Sincerely,

Richard Turner PhD

Consulting editor, PLOS Medicine

plosmedicine@plos.org

In the text and supplementary material, several organizations are mentioned in the context of provision or packaging of study drugs (e.g., "Accord Healthcare" and "Piramal"). We feel that all such contributions should mentioned in the financial disclosures (submission form). 

We ask you to adapt the competing interest statement (submission form) to name relevant members of PLOS Medicine's editorial board among the authors. 

We suggest adapting the title as follows: "Evaluation of Hydroxychloroquine or chloroquine for prevention of COVID-19 (COPCOV): A double-blind, randomised, placebo-controlled trial

Please adapt the short title to contain brief details of the study. 

Please ensure that the revised abstract follows CONSORT; if secondary endpoint outcomes are to be reported, all should be included to avoid an appearance of cherry-picking. 

Please combine the "Methods" and "Findings" subsections of the abstract. 

Please add a new final sentence to the combined subsection, which should begin "Study imitations include ..." or similar and should quote 2-3 of the study's main limitations. 

Please quote aggregate demographic details for study participants in the abstract.

In place of the "Research in Context" section, please substitute a new "Author Summary" section in PLOS Medicine style. You may find it helpful to consult one or two recent research papers in the journal to get a sense of the preferred style. 

Noting comments from the referees on the statement "Published studies suggest clinical benefit", please adapt the abstract and main text to reflect the implications of the published individual studies. 

Please adapt the final sentence of the Introduction (main text; "We present the results ...") to state the study's aim.

Please restructure the Discussion section (main text) so that the first paragraph summarizes the study's main findings, with these being discussed in subsequent paragraphs.

Please remove all trademarks from the paper. 

Please remove the information on funding from the title page and from the end of the main text. In the event of publication, this information will appear in the article metadata, via entries in the submission form. 

In the reference list, please convert italics to plain text. 

Please add accessed dates to online references such as reference 1.

Noting reference 2, please list 6 author names followed by "et al.", where appropriate.

Please use the journal name abbreviation "PLoS ONE".

Please adapt the title for Fig 1 to "Participant flowchart" or similar. 

Please include a completed CONSORT checklist as a supplementary file, labelled "S1_CONSORT_Checklist" or similar and referred to as such in the Methods section (main text). 

In the checklist, please refer to individual items by section (e.g., "Methods") and paragraph number, not by line or page numbers as these generally change in the event of publication. 

Is there a formal study protocol that you can attach as "S2_Protocol"?

Please rename the attached statistical analysis plan "S3_Statistical Analysis Plan" or similar. 

Comments from the reviewers:

*** Reviewer #1: 

Statistical review

This paper reports a RCT comparing pooled Hydroxychloroquine + chloroquine vs placebo for prevention of covid-19 infection. Although the sample size enrolled was much lower than target, the trial is still very large and provides useful evidence about the positive effect of the drugs. The trial was reported well and I had only some minor comments:

1. Abstract "The primary endpoint was symptomatic COVID-19": perhaps add 'during the follow-up period.' to clarify that this wasn't time-to-event.

2. Abstract: The secondary outcomes reported in the abstract do not align perfectly with the secondary outcomes in the clinicaltrials.gov registration page (or later in the paper). If those outcomes were collected it would be good to report them in the abstract.

3. Abstract: "Clinical trials need to be supported and protected, particularly in pandemic emergencies." - Although I see the rationale for this statement after reading the paper's discussion, I didn't see how this followed from the rest of the abstract so would leave it out.

4. Page 9: were any covariates adjusted for in the log-binomial models?

5. Page 9: was there non-negligible loss-to-follow-up (the sample size calculation planned for 20%) and if so how was this handled in the ITT analysis?

6. Page 10 "In symptomatic patients there were no differences in symptom severity scores." - I would add no significant difference (and preferably would include the median score in each arm, similarly to other outcomes have summary by arm).

7. Page 10 "a mean difference of 104 days (95% CI, 12 to 199)." - is there a reason the authors did not provide a p-value for this outcome? Other secondary outcomes have them.

James Wason

*** Reviewer #2: 

In this double-blind, placebo controlled, multi-national, randomised trial the authors test whether a three month chemoprophylaxis with hydroxychloroquine or chloroquine can reduce the incidence of clinical COVID-19. Several secondary outcomes were also reported

The trial was planned for 40,000 participants, of whom 4,652 were recruited, after which point the trial had to close due to lack of recruitment. 

The hypothesis is clearly stated, and, despite the premature closure, I believe, within reasonable clinical certainty, can conclude on the hypothesis. 

The results seem clear, and the wording in the interpretation, are well-balanced by the authors. They also manage to balance the limitations. 

In all aspects, apart from the premature closure, this is a very high quality RCT. 

I strongly agree with the authors on their point that especially during crises like the COVID-19 pandemic, stakeholders should stand up for the completion of RCT´s since, at the end of the day, this design is the only that can inform us solidly, when conducted rigorously. 

I have few comments: 

The authors state :

"This allowed for approximately 20% loss to follow up, withdrawals, protocol deviations and non-adherence, and provided 80% power to detect a 23% reduction in symptomatic COVID-19 incidence (from an estimated 3%) for each drug individually with 95% confidence. There were protracted delays and difficulties with recruitment as the study was starting in 2020"

This seems extremely ambitious, however, sometime the "perfect becomes enemy of the good", insofar as the chance of completing extremely ambitious projects decreases with the increasing target sample size. I speak of bitter personal experience…. As far as I can see, the sample size calculation wanted to detect a difference between approx. 3% to approx. 2.4%. In light of the pandemic threat and the situation when the trial was planned, I am a little puzzled: would a protective effect as small as this actually have been clinically relevant in stopping the pandemic? Could the authors elaborate on this in the discussion

In the discussion, first part: "Few drugs have excited such controversy as hydroxychloroquine in COVID-19. Bureaucracy, politicisation, unfounded opinions, false claims and outright fraud conspired against the conduct of the clinical trials needed to provide objective evidence early in the pandemic" 

I agree, this was outrageous. However, I feel, the results of the trial should be respected in the first lines of the discussion. This is an opinion, I´m aware, however, maybe the authors should "kill this darling", at least regarding the prioritizing of messages in the discussion. Maybe such a statement (that may for some readers be controversial) should be placed less prominently. This is said by a trialist who had a very public controversy with my national medicines agency (for closing down a HC trial during the first wave) so I do feel much the same as in this statement, nevertheless, maybe only a minority of the readers think this is key ? Just a comment, I guess the authors have had some relevant discussions on this…

Further: 

"This trial provides no support for these claims, and it supports the generally good short-term safety profile of these drugs.30 It challenges the World Health Organization "living" guidelines on COVID-19 chemoprevention which emphasised toxicity in COPCOV Lancet V2 Page 12 of 26 recommending against hydroxychloroquine. The same guideline also stated that prophylactic use "probably has a small or no effect on laboratory-confirmed COVID-19" with "moderate certainty" , and has no effect on mortality with "high certainty".26 Yet no deaths were reported in any of the pre-exposure prophylaxis studies, while the drugs have been well-tolerated with very low rates of treatment discontinuation (Figures S5, S6).31"

This is extremely important, which I strongly agree on, and actually it raised serious concern on the WHO handling of the pandemic on a very key point. I do not think the aftermath in WHO has handled this erroneous and very harmful reaction at a crucial point in the pandemic in a relevant way. 

Later: "Now, in early 2023, with increasing coverage of effective vaccines," - possibly change to "Now, with increasing…" to strengthen generalizability of the statement

Another point. I think the discussion lacks a discussion of the interesting and very relevant finding of a reduction in the incidence of respiratory infections. Focus on this would make the results relevant in a future context, it indicates a general effect in respiratory infections. 

Limitations are very well discussed. 

Finally, I want the congratulate the authors with a very well-planned, well conducted RCT, despite of the unreasonable resistance and irrelevant political decisions that made life very hard for people who really wanted to inform the world with precise data, on important issues during the COVID-19 pandemic. 

*** Reviewer #3: 

The authors present the results of the long-awaited COPCOV study, which evaluated chloroquines as chemoprophylaxis for COVID-19. This may sound outdated now that we fight the pandemic with vaccines, but there were times when vaccines did not exist and the repurpose of existing drug was the best option to tackle the pandemic. Its results will not have a direct impact on the management of the current pandemic, but will help addressing challenges that caught the scientific community off-guard when the pandemic hit in 2020. The design and implementation of the study are sound

Major comments

- Figure 2 should contain the primary outcome. 

- Please refer to the primary outcome as "symptomatic, laboratory-confirmed COVID-19" throughout the manuscript. That terminology is used in methods when the primary outcome is described, but not in the results section, which is confusing. 

- I find the "primary outcome" section in results confusing:

 o It starts reporting SARS-CoV-2 infection, which is not a primary outcome (it is actually an outcome that is not described in methods). Given that PCR tests were not done systematically, how is SARS-CoV-2 infection different from "symptomatic covid-19"? This needs to be clarified in methods. 

 o Then it reports symptomatic [laboratory-confirmed] covid-19 which is the primary outcome

 o And then it reports comparisons that of non-primary outcomes: pcr-confirmed symptomatic covid-19, symptomatic covid-19 diagnosed by seroconversion, DBS symptomatic COVID-19. 

 o So, if you read this section fast (as most of us do), it is hard to identify the main result of the study. I think some rearrangement can make this section clearer. 

* Describe time to vaccination (KM curve) by arm. How were these patients handled?

* Make sure that the KM estimations are truly an ITT analysis, i.e., that patients who discontinue the drug are not censored (which is what the SAP says in page 14, but that is not an ITT analysis)

* It is odd that the main analysis is not a time-to-event analysis (e.g. by using a Kaplan Meier estimator). This means that the authors assume there is not right-censoring, when in fact there is according to Figure S7. I don't think that the results will change substantially, but I think a time-to-event analysis should be provided too (maybe as supplemental).

* I think a KM curve for the main outcome with the following three arms should be added (maybe as supplemental): control, chloroquine, hydroxychloroquine. 

* Effect estimates in an absolute scale (i.e., risk differences) should be provided.

Regarding the meta-analysis:

* Addressing the risk of bias of the component studies is standard practice in meta-analyses. See, for example Supp Table 1 of Garcia-Albeniz et al. I think it would be good to use the Cochrane risk of bias 2.0 tool to address the risk of bias of those studies not already evaluated for bias in the study by Garcia-Albeniz et al. 

* Figure 3:

 o The confidence interval by Seet et al does not match the original publication (0.44-0.97)

 o The estimate by Polo et al should be 0.49 (0.00-2.29)

 o I would not include the article by McKinnon et al in this meta-analysis because of the same reason that Garcia-Albeniz (reference 23) used: although the authors do not label the study as post-exposure prophylaxis, they report that "60% [of study participants] reported contact with a COVID-19 positive patient before study entry

 o The estimate by Vijayaraghavan should be 0.85, 95% CI 0.35 to 2.07

 o The upper limit by Llanos-Cuentas should be 7.11

 o For consistency with Garcia-Albeniz et al, I think the authors should estimate the confidence interval of the meta-analyzed risk ratio using both a fixed (common) effect model and a random effects model. 

 o Figure 3 should have a note explaining the meaning of the grey squares around the point estimates

 o I think it would be good to complement the meta-analysis with the risk difference estimates, similar to Supp Figure 1 of Garcia-Albeniz et al.

* Figure S5

 o The estimate by Polo et al should be 0.49 (0.00-2.29)

Minor comments:

- If the sample size allows for an informative analysis, it would be nice to see a subgroup analysis of only healthcare workers

- Results from the drug measurement described in the sixth line of "Procedures" section are not reported

- I don't see the need to report p-values or concluding that "there were/were not statistically significant differences" because I consider this an outdated practice. https://www.nature.com/articles/d41586-019-00857-9

- There is a section "Interpretation of results" in the supplementary materials, that is inside the meta-analysis section and it reads like it does not belong there. 

*** Reviewer #4: 

This randomized controlled trial investigated the effect of using hydroxychloroquine (HCQ) as a preventive tool against symptomatic covid-19 (main endpoint). This study found that there is a borderline difference in the incidence of covid-19 RR=0.85 (0.72-1.00), p=0.05 and there was no effect against asymptomatic COVID-19 (RR: 0.96 (95% CI, 0.82 to 1.12), p=0.6) nor for all SARS-COV-2 infections in ITT analyses. In per protocol analysis, the primary outcome showed no effect.

The strengths of this study include the larger number of participants included (4600) compared to other clinical trials on hydroxychloroquine as a preventive medication. The trial took place in multiple countries.

My main concern is that the preventive effect is only observed among confirmed SAS-CoV-2 PCR (with a small number of cases confirmed by PCR) but there is no effect when the diagnosis is based on seroconversion or based on all SARS-CoV-2 cases or only asymptomatic cases. These drugs may have a small or no effect on COVID-19. 29% (13-14% loss to follow-up) of the initial randomized population have been excluded from per protocol analyses. Additionally, timing of this study may may no longer be as important now that vaccines are available and the variants of SARS-CoV-2 are different.

The description and the methods used in the meta-analysis raise some questions.

Main analysis:

1) "The study took place in multiple countries, including the United Kingdom, Asia, and Africa." Is there a geographical effect? Standard of cares, exposure to the SRAS-CoV-2 (according to preventive policies), population are variable. It could be interesting to have analyses per continent or broad geographic areas

2) "The recruitment for the study took place between April 2020 and March 2022. Various variants, such as Alpha and Delta of the SARS-CoV-2, emerged during this period. Are these differences being considered? Regarding the vaccines, their effectiveness varied depending on the variants."

3) "The analyses combine two different molecules (CQ and HCQ). Is there a difference in their effects?"

4) "How was adherence to the intervention assessed? Plasma concentrations of HCQ or CQ were not measured."

5) "The power calculation was originally presented for a total of 40,000 participants. However, only 4,652 people were ultimately randomized. What is the expected detectable effect size?

6) The authors have longitudinal data with almost 2 years of follow-up. Why is the primary analysis based on a Fisher exact test? I wonder if the results would be different with a survival model and Kaplan-Meier analysis (Kaplan-Meier was performed for respiratory infections but not for COVID-19). The main result for the primary outcome is borderline significant (RR=0.85 [0.72-1.00], p=0.05), it is difficult to interpret this result. Results for all SARS-CoV-2 infection suggest no effect of CQ and CHQ.

7) Table 2 mentions missing data for the primary outcome. What was the number of missing data? This should be specified.

8) In the discussion, statement is not correct "This protective effect is consistent with previous studies10,11,14-22" I looked at the first references mentioned: ref 10, Syed et al. 2021 writes "It is concluded that the PrEP HCQ does not significantly prevent COVID-19 among high-risk HCPs." Abella et al. 2021 wrote "there was no clinical benefit of hydroxychloroquine administered daily for 8 weeks as pre-exposure prophylaxis in hospital-based HCWs exposed to patients with COVID-19." Same for Rajasingham et al. 2021: Pre-exposure prophylaxis with hydroxychloroquine once or twice weekly did not significantly reduce laboratory-confirmed COVID-19 or COVID-19-compatible illness among healthcare workers.

9) In the Abstract, « Published studies suggest clinical benefit" is not right. Several randomized trials have shown no effect of HCQ on covid19 as treatment or preventive tool… These results could be compared with negative results from previous meta-analysis studying HCQ as preventive tool (Hong et al. 2023). Previous meta-analyses of RCTs on HCQ as treatment also reported no effect on COVID-19.

Hong H, Friedland A, Hu M, Anstrom KJ, Halabi S, McKinnon JE, Amaravadi R, Rojas-Serrano J, Abella BS, Portillo-Vázquez AM, Woods CW, Hernandez AF, Boulware DR, Naggie S, Rajasingham R. Safety and efficacy of hydroxychloroquine as prophylactic against COVID-19 in healthcare workers: a meta-analysis of randomised clinical trials. BMJ Open. 2023 Jun 16;13(6):e065305. doi: 10.1136/bmjopen-2022-065305. PMID: 37328184; PMCID: PMC10276967.

10) I wonder whether the protocol was pre-registered on an open-platform. 

Meta-analysis: The meta-analysis suggests a protective effect. It is important to underline that this study has a weight of 61%.

 it is unclear what RR (or Odds Ratio or Hazard Ratios ; adjusted or not) was used. A table describing the individual included studies could be useful to understand what was the intervention, the dose, the placebo, the duration, and the endpoint, study population (general population vs health workers).

1) Heterogeneity could be added in the main text

2) In the supplementary material, it is not clear what meta-analysis model has been used: fixed or random effect model?

3) This meta-analysis pooled different interventions: I looked at the studies with the most important weight in the meta-analysis. The meta-analysis pooled different endpoints across the studies (PCR positive or probable covid-19, symptomatic covid only, infections…). I have not checked all the included studies but some of them included a different placebo (Vitamin C)

4) In Rajasingham 2020, the Hazard ratio is .74 (95% CI, 0.45-1.20) for COVID-19 compatible with symptoms and the forest plot in this manuscript reported slightly different values 0.74 (95% CI, 0.50-1.10). Placebo was Vitamin C.

5) In Seet 2022 trial, the placebo was Vitamin C. The value used in the forest plot is 0.70 [0.47 ; 1.04]. Was it recalculated for symptomatic COVID-19? In the original study of Seet, Odds Ratio are provided and not Relative Risk. And the aOR is different: 0.70 [0.44 ; 0.97] for SARS-CoV2-infection. 

6) I also wonder whether there is a dose-effect of HCQ/CQ (using meta-regression)

I suggest to add line numbering in the .pdf

To conclude, I wonder to what extent a prevention policy for SARS-CoV-2 using hydroxychloroquine can be implemented and useful now that vaccines are available (as it is mentioned in the discussion). The recommendation would be to advise the general population to take hydroxychloroquine regularly every day. Is this sustainable and feasible? Unlike vaccines, which have shown a low preventive effect, where we only need one injection and boosters.

***

[LINK]

---

## [Decision Letter · Decision Letter 2]

26 Nov 2023

Dear Dr. Schilling,

Thank you very much for submitting your revised manuscript "Evaluation of hydroxychloroquine or chloroquine for prevention of COVID-19 (COPCOV): a double-blind, randomised, placebo-controlled trial" (PMEDICINE-D-23-02452R2) for consideration at PLOS Medicine. 

Your paper was discussed among the editors and re-seen by three of our independent reviewers, including the statistical reviewer. The reviews are appended at the bottom of this email and any accompanying reviewer attachments can be seen via the link below:

[LINK]

We will not be able to accept the manuscript for publication in the journal in its current form, but would like to invite you to submit a further revised version that addresses the editors' comments fully. Again, we will be unable to make a decision about publication until we have seen the revised manuscript and your response, and we may seek re-review by one or more of the reviewers. 

We hope to receive your revised manuscript by Dec 11 2023 11:59PM. Please email us (plosmedicine@plos.org) if you have any questions or concerns.

Please let me know if you have any questions, and we look forward to receiving your revised manuscript. 

Sincerely,

Richard Turner PhD

Consulting editor, PLOS Medicine

plosmedicine@plos.org

Please remove the information on funding and the publishing licence from the abstract page. In the event of publication, information on funding will appear in the article metadata, via entries in the submission form. 

Around line 87 (abstract), please add a sentence to mention the intended trial recruitment and explain briefly why it could not be achieved. 

At lines 94, 95, 96 & 99 (abstract), we suggest substituting colons for the four semicolons. 

Please remove the sentence at line 107 ("Difficulties conducting ...", abstract), as this issue can be fully dealt with in the Discussion (main text). 

Please reformat the 'Author summary' so that the three subsections each comprise 3-4 short points of 1-2 sentences each. Please use the active voice (e.g., "We found ...") in one or two points. 

Regarding the summary of outcomes at line 218, please include a full list of all outcomes. We note that there are 'other outcomes' specified, including impact on healthcare costs. We ask you to report all outcomes for which data are available in the present paper. 

At line 325, noting 'preventitive', we suggest using 'preventive' throughout.

Please adapt the reference call-outs to journal style throughout, e.g., "... equivalent [25,26]." (noting the absence of spaces within the square brackets). 

Despite the retraction, please complete the citation for reference 4 with full access details. Please revisit reference 24, which may be missing some information. 

Comments from the reviewers:

*** Reviewer #1: 

Thank you to the authors for addressing my previous comments well, I have no further issues to raise.

*** Reviewer #2: 

I have no further comments, the authors have responded very adequately. It´s a fine study

*** Reviewer #3: 

All my comments have been addressed

***

[LINK]

---

## [Editor Report · Decision Letter 3]

24 Dec 2023

Dear Dr. Schilling,

Thank you very much for submitting your revised manuscript "Evaluation of hydroxychloroquine or chloroquine for prevention of COVID-19 (COPCOV): a double-blind, randomised, placebo-controlled trial" (PMEDICINE-D-23-02452R3) for consideration at PLOS Medicine. 

Your revised paper was evaluated by our academic editor and discussed among the editorial team. In light of our discussions we will not be able to accept the manuscript for publication in its current form, but we would like to again invite you to submit a revised version that fully addresses the remaining comments; we may seek re-review by one or more of the reviewers. 

In revising the manuscript for further consideration, your revisions should address the specific points made by the editors. Please also check the guidelines for revised papers at http://journals.plos.org/plosmedicine/s/revising-your-manuscript for any that apply to your paper. In your rebuttal letter you should indicate your response to the reviewers' and editors' comments, the changes you have made in the manuscript, and include either an excerpt of the revised text or the location (eg: page and line number) where each change can be found. Please submit a clean version of the paper as the main article file; a version with changes marked should be uploaded as a marked up manuscript.

We hope to receive your revised manuscript early in the new year. Please email us (plosmedicine@plos.org) if you have any questions or concerns.

Please let me know if you have any questions, and we look forward to receiving your re-revised manuscript. 

Sincerely,

Richard Turner PhD

Consulting editor, PLOS Medicine

plosmedicine@plos.org

In the 'Financial disclosure' section (submission form), the information about the CC-BY licence is superfluous and can be removed, as this licence is standard at PLOS Medicine. 

Please remove the quoted secondary outcome findings from the abstract, retaining the primary outcome findings, results of the prespecified meta-analysis and information about safety. 

At line 152 (Introduction), please adapt the wording to "a false claim".

At line 163 (Introduction) please adapt the text to state that the trials did not individually provide evidence of benefit. 

Noting the information around line 288 (Results) regarding confirmed COVID-19 infection, we ask you to add a supplementary table listing causes of respiratory illness. 

At line 330 (Discussion) we feel that "... evidence of moderate preventive efficacy" is an overstatement and ask you to amend the wording to "... some evidence of preventive efficacy" or similar. 

At line 331, we ask you to adapt the text to "... an apparent 15% reduction", noting the equivocal p value. 

At line 332 we suggest "efficacy" rather than "effect", as elsewhere in the text. 

Again at line 332, we ask you to revisit "protective effect is consistent with previous smaller studies". We believe that the previous studies did not provide evidence of significant benefit, and we ask you to adapt the wording to indicate this clearly.

At line 360, we suggest removing "outright fraud", although we are aware that this is not an unfounded statement. 

At line 374, we ask you to remove the sentence "This must not be allowed ...". 

At line 389, we ask you to remove the word "strong", which seems an overstatement based on the numbers quoted. 

At line 397, please remove the phrase "and not impeded". 

At line 403, please remove the information about study funding, which is present in the financial disclosures which will appear in the article metadata in the event of publication. 

Please edit reference 4: we believe that this can be truncated prior to "Epub ahead of print" so as to retain the essential information. 

***

---

## [Editor Report · Decision Letter 4]

15 Feb 2024

Dear Dr. Schilling,

Thank you very much for submitting your revised manuscript "Evaluation of hydroxychloroquine or chloroquine for prevention of COVID-19 (COPCOV): A double-blind, randomised, placebo-controlled trial" (PMEDICINE-D-23-02452R4) for consideration at PLOS Medicine. 

Following from the recent email exchange, your provisional revised text was discussed among the editorial team, yielding our further specific requests (below). We would now like to invite you to submit a further revised version for consideration in which these points have been fully addressed. In your rebuttal letter you should indicate your response to the reviewers' and editors' comments, the changes you have made in the manuscript, and include either an excerpt of the revised text or the location (eg: page and line number) where each change can be found. Please submit a clean version of the paper as the main article file; a version with changes marked should be uploaded as a marked up manuscript.

We hope to receive your revised manuscript within one week. Please email us (plosmedicine@plos.org) if you have any questions or concerns.

We suggest a further email exchange if you would like to discuss any specific point(s) prior to resubmission, and otherwise look forward to receiving your revised manuscript. 

Sincerely,

Richard Turner PhD, 

Consulting editor, PLOS Medicine

plosmedicine@plos.org

1. Line 78: We suggest abbreviating HCQ here (rather than at line 83). 

2. Line 79: Please adapt the text to “Previous randomised controlled trials did not show benefit of HCQ against COVID-19 and, although meta-analysis suggested clinical benefit, guidelines recommend against its use." 

3. Line 86: Please list the secondary and tertiary outcomes after the primary endpoint, adding “(not reported here)” where appropriate. 

4. Line 88: Please adapt the text to “… because of protracted delays owing to controversies regarding efficacy and adverse events with HCQ use, vaccine rollout in some countries, and other factors. Between 29 April 2020 …”. 

5. Line 94: We ask you to adapt the text to “… was similar in the HCQ/CQ compared with the placebo arms: ...”. 

6. At line 97, please move the p value inside the brackets for consistency. 

7. At line 101, please remove “pre-exposure” (redundant given “prophylaxis” later in the sentence). 

8. At line 103, please remove “very” (prior to “well tolerated”), which we feel is unnecessary.

9. Line 109: please adapt the text to “In this large placebo-controlled, double-blind randomised trial, HCQ and CQ were safe and well tolerated in COVID-19 chemoprevention, and there was evidence of moderate protective benefit in a meta-analysis of evidence from this trial and similar RCTs.”

10. At line 118, please adapt the text to: “HCQ proved ineffective in the treatment of hospitalised patients, and individual RCTs testing COVID-19 prophylaxis did not show benefit of HCQ. However, a meta-analysis of trial data suggested some efficacy in preventing COVID-19.”

11. At line 121, please remove “In contrast”.

12. At line 129, please adapt the text to: “We found that HCQ and CQ were well tolerated and safe in prophylaxis. There was some evidence for protection against symptomatic COVID-19, and an apparent reduction in workdays lost to illness.”

13. At line 147, please remove “verging on panic”. You may wish to adapt the sentence to “… global concern about the projected consequences of the developing pandemic for large-scale human ill-health and mortality” or similar. 

14. At line 149, please define the abbreviations “CQ” and “HCQ”, assuming that you plan to use these throughout the text. 

15. At line 153, please remove reference 4 and instead cite https://www.nature.com/articles/d41586-020-01695-w or a similar article discussing the situation. 

16. At line 154, we ask you to amend the text to “… regulatory decisions.”

17. At line 160, regarding “… recommended widely”, please add a reference as we are not clear what kind of recommendations are being discussed. 

18. At line 162, please add a comma after “Despite this”.

19. At line 164/5, please run the sentence on (“… underpowered to demonstrate benefit [23], but the evidence ...”.

20. Regarding the presentation around line 220, we note that the clinicaltrials.gov page (https://classic.clinicaltrials.gov/ct2/show/NCT04303507?term=copcov&draw=2&rank=1) currently lists one primary outcome, four secondary outcomes and four “other outcomes”. We suggest editing the page to match the definitive version of the trial protocol. 

21. After line 230, please list the endpoints in continuous text rather than bulleted text.

22. At line 336, please report the RR and 95% CI (as at line 102) rather than “reduced by 20%” and the like. 

23. Line 344: We ask you to adapt the text to “… was able to provide evidence relevant to possible protective efficacy. For the trial’s primary endpoint (incidence of symptomatic laboratory-confirmed COVID-19), the non-significant finding in favour of HCQ/CQ treatment is consistent with the aggregated results of previous smaller studies [10,11,14-22], all but one of which had a lower incidence of symptomatic COVID-19 in the HCQ treatment arm (Figure 3). Incorporating the COPCOV trial data in a prespecified meta-analysis of pre-exposure RCTs, with little heterogeneity across the studies and no evidence of publication bias, suggests a moderate protective benefit (RR 0.80, 95% CI 0.71-0.91).”

24. At line 351, please remove “41%”. 

25. At line 352, please remove “approximate 25%”.

26. Line 372: do you mean the meta-analysis reported in this paper? Please adapt the wording to clarify if so. 

27. At line 375, we ask you to adapt the wording to “Bureaucracy, politicisation and heated debate obstructed the undertaking of clinical trials needed to provide objective evidence early in the pandemic.”

28. At line 385, please adapt the text to “… despite the seriousness of the pandemic, there were major obstacles to the conduct of urgently needed clinical research, particularly in low-resource settings.”.

29. At line 408, please adapt the text to “… was safe and well-tolerated, and combined with data from other similar trials provided some evidence that laboratory-confirmed symptomatic COVID-19 might be reduced.”

30. Please reformat table 2 to indicate the ranking of the endpoints. 

***

---

## [Editor Report · Decision Letter 5]

23 May 2024

Dear Dr. Schilling,

Thank you very much for re-submitting your manuscript "Evaluation of hydroxychloroquine or chloroquine for prevention of COVID-19 (COPCOV): a double-blind, randomised, placebo-controlled trial" (PMEDICINE-D-23-02452R5) for consideration at PLOS Medicine.

With apologies for the tardy reply, I have discussed the paper with editorial colleagues and I am pleased to tell you that, provided the remaining editorial and production issues are fully dealt with, we expect to be able to accept the paper for publication in the journal.

[LINK]

Please let me know if you have any questions in the meantime, and we look forward to receiving the revised manuscript.   

Sincerely,

Richard Turner, PhD

Consulting Editor, PLOS Medicine

plosmedicine@plos.org

Requests from Editors:

Should that be "HCQ" at line 85 (abstract)?

At line 107, please make that "... provided a moderate protective benefit ...". 

At line 121 (author summary), please indicate the abbreviations for CQ and HCQ and use these abbreviations thereafter in the author summary. 

At line 164, alongside "to indicate a lack of efficacy", do you mean for CQ/HCQ or for the various agents mentioned earlier in the sentence more generally? You may wish to add a few words to clarify. 

Again in the introduction (main text) HCQ can be used from line 161 onwards. 

Please include a completed CONSERVE checklist as a supplementary file, if relevant. 

***

---

## [Editor Report · Decision Letter 6]

14 Jun 2024

Dear Dr Schilling, 

On behalf of my colleagues, I am pleased to inform you that we have agreed to publish your manuscript "Evaluation of hydroxychloroquine or chloroquine for the prevention of COVID-19 (COPCOV): a double-blind, randomised, placebo-controlled trial" (PMEDICINE-D-23-02452R6) in PLOS Medicine.

PRESS

Sincerely, 

Richard Turner, PhD 

Consulting Editor, PLOS Medicine

plosmedicine@plos.org